# A double role of the Gal80 N terminus in activation of transcription by Gal4p

Annekathrin Reinhardt-Tews[1], Rościsław Krutyhołowa[2,3], Christian Günzel[1], Constance Roehl[1], Sebastian Glatt[2], Karin D Breunig[1]

**The yeast galactose switch operated by the Gal4p–Gal80p–Gal3p regulatory module is a textbook model of transcription regulation in eukaryotes. The Gal80 protein inhibits Gal4p-mediated transcription activation by binding to the transcription activation domain. In *Saccharomyces cerevisiae*, inhibition is relieved by formation of an alternative Gal80–Gal3 complex. In yeasts lacking a Gal3p ortholog, such as *Kluyveromyces lactis*, the Gal1 protein (KlGal1p) combines regulatory and enzymatic activity. The data presented here reveal a yet unknown role of the KlGal80 N terminus in the mechanism of Gal4p activation. The N terminus contains an NLS, which is responsible for nuclear accumulation of KlGal80p and KlGal1p and for KlGal80p-mediated galactokinase inhibition. Herein, we present a model where the N terminus of KlGal80p reaches the catalytic center of KlGal1p causing enzyme inhibition in the nucleus and stabilization of the KlGal1–KlGal80p complex. We corroborate this model by genetic analyses and structural modelling and provide a rationale for the divergent evolution of the mechanism activating Gal4p.**

## Introduction

Transcription regulation in response to the environment occurs in all organisms and is essential for cellular homeostasis. Studies on the response to changing nutrients in microorganisms have provided deep insight into the adaptive processes at the molecular level. Pioneering work on the regulation of transcription by galactose in yeast gave rise to the Douglas–Hawthorne model of the *GAL* regulon (Douglas & Hawthorne, 1972; Oshima, 1991). This model assumes that a gene-specific transcription activator (Gal4p) is inhibited by Gal80p. When a yeast cell has access to galactose the Gal80p-mediated inhibition is relieved by the Gal3p protein, which functions as a galactose transducer.

These three regulatory proteins, namely Gal4p, Gal80p, and Gal1/3p, represent the heart of the "galactose switch." Gal4p transmits the information leading to high transcription rates to the transcription machinery. Gal80p mediates inhibition by binding with high affinity to a specific site overlapping the C-terminal activation domain of Gal4p (Keegan et al, 1986; Ma & Ptashne, 1987; Ansari et al, 1998). Gal1p, the enzyme that catalyzes galactose phosphorylation by ATP, senses the presence of galactose intracellularly by binding galactose and ATP directly, which increases the affinity between Gal1p and Gal80p. Therefore, insights into the formation of the Gal1–Gal80 complex are central to the understanding of the *GAL* switch and the activation of Gal4p. The above described trimeric regulatory module is present in *Kluyveromyces lactis* (Zenke et al, 1993, 1996), whereas *Saccharomyces cerevisiae* harbors a more complex *GAL* switch (Johnston, 1987) As a result of a genome duplication event in the *Saccharomyces* lineage a paralog of Gal1p exists, named Gal3p. The Gal3p variant of Gal1p has lost enzymatic activity and seems to be dedicated to regulating Gal4p activation by formation of the Gal3–Gal80 complex (Hittinger & Carroll, 2007). The ScGal1p variant has only residual regulatory activity because its affinity to ScGal80p is lower than that of Gal3p (Lavy et al, 2016).

Gal4p has been shown to work as a transactivator, which is inhibited by Gal80p, in many types of higher eukaryotic cells stressing the evolutionary conservation of the transcription apparatus. Multiple studies dissected in great detail how Gal4p and Gal80p regulate gene expression and the regulation of the *GAL* genes in yeast became a textbook model for eukaryotic gene regulation. Nevertheless, there are still many open questions concerning the molecular details of its regulation. Specifically, the mechanics of Gal4p activation is poorly understood.

In the *Kluyveromyces* lineage, which in contrast to *S. cerevisiae* has not undergone genome duplication, KlGal1p is a so-called "moonlighting" protein, which has two distinct functions, the enzymatic and the Gal3-like regulatory function. These functions can be separated by specific mutation (Meyer et al, 1991; Zenke et al, 1996; Vollenbroich et al, 1999; Hittinger and Carroll, 2007). Cross-complementation experiments have indicated and crystal structures of KlGal80p,

[1]Institut für Biologie, Martin-Luther-Universität Halle-Wittenberg, Halle (Saale), Germany  [2]Malopolska Centre of Biotechnology, Jagiellonian University, Krakow, Poland  [3]Faculty of Biochemistry, Biophysics and Biotechnology, Jagiellonian University, Krakow, Poland

Correspondence: karin.breunig@serymun.com; sebastian.glatt@uj.edu.pl
Karin D Breunig's present address is serYmun Yeast GmbH, Halle, Germany

ScGal80p, as well as ScGal1p and ScGal3p have confirmed a high conservation of the Gal4p-Gal80p-Gal1/3 modules (Zenke et al, 1996; Menezes et al, 2003; Thoden et al, 2007, 2008; Lavy et al, 2012). However, several differences have been documented. (i) *ScGAL3* is unable to provide the *Klgal1* regulatory function in *K. lactis* unless *KlGAL80* is replaced by *ScGAL80*, indicating some incompatibility between ScGal3p and KlGal80p (Zenke et al, 1996). (ii) KlGal80p is a nuclear protein whereas ScGal80p is nucleocytoplasmic (Anders et al, 2006b). (iii) Species-specific phosphorylation of Gal80p and Gal4p have been reported (Mylin et al, 1989, 1990; Zenke et al, 1999), but the role of post-translational modifications in Gal4p regulation is poorly understood. Therefore, it is not entirely clear if the mechanistic details of the *GAL* switch in *K. lactis* and *S. cerevisiae* are identical or not.

Here, we have addressed species specificity by identifying the NLS of KlGal80p. We show that the extreme N terminus of the KlGal80 protein contains a NLS responsible for nuclear import of KlGal80p and involved in formation of the KlGal80–KlGal1 complex. Both functions are overlapping and important for KlGal4p regulation in *K. lactis*. We present evidence for a so far unknown direct interaction between the KlGal80 N terminus and the catalytic center of KlGal1p explaining the KlGal80p-mediated inhibition of KlGal1's galactokinase activity and the influence of KlGal80p on KlGal4p activation by galactose. In addition, we present a structural model of the complex based on the atomic structures of free KlGal80p and of the ScGal3–ScGal80 co-complex. Furthermore, we discuss that despite strong sequence similarity the functions of the KlGal80p N terminus are not conserved in *S. cerevisiae* in the light of the divergent evolution of the two yeast genera.

# Results

### Nuclear localization of Gal80p in *K. lactis* depends on the N terminus

The Gal80 protein of *K. lactis* (KlGal80p) differs in its subcellular distribution from its homolog in *S. cerevisiae*. Whereas ScGal80p is localized both, in nucleus and cytoplasm (nucleocytoplasmic), KlGal80p is exclusively nuclear (Anders, 2006a; Anders et al, 2006b). To identify the NLS various sub-fragments of KlGal80p were fused to GFP and constitutively expressed in a *Klgal80* deletion mutant. Localization of the fusion proteins was analyzed by fluorescence microscopy (Fig S1). Besides full-length KlGal80p (Fig 1A), all protein segments conferring nuclear localization contained the N terminus (Figs S1 and 1). The smallest nuclear segment comprised only the first 39 amino acids (KlGal80-C1p, Fig 1B). GFP-KlGal80p with a deletion of those N-terminal amino acids (KlGal80-DC1p) was found in the whole cell (Fig 1C). Taken together, these data clearly indicate that an NLS is located at the N terminus of KlGal80p.

Indeed, the KlGal80 N terminus contains a sequence rich in basic residues $^5$KRSK$^8$, which resembles a monopartite, class 2 NLS consensus sequence, K-K/R-X-K/R (Chelsky et al, 1989) (Fig 2, top). To test the functionality of this sequence two of the basic residues, namely, lysine 5 and arginine 6 were mutated (K5A, R6A) by site-directed mutagenesis of the reading frame of GFP-KlGal80-C1p (aa

2–39). Like the N-terminal deletion, this point mutation abolished nuclear accumulation and showed nucleocytoplasmic localization (Fig 1D). To rule out an effect of overexpression of the GFP-KlGal80-K5A/R6A protein, which was encoded by a multicopy plasmid, the mutation was also introduced into the chromosomal *KlGAL80* gene. Comparing the ("wild-type") parent strain JA6/G80M encoding a (c-myc)$_3$-tagged variant of KlGal80p with the resulting mutated strain JA6/G80-KR56A (*Klgal80-K5A/R6A*) by immunofluorescence microscopy confirmed that the basic residues K5 and R6 are important for nuclear localization. KlGal80-(c-myc) p was nuclear under inducing conditions as shown before (Anders et al, 2006b), whereas KlGal80-K5A/R6A-(c-myc)$_3$p was nucleocytoplasmic (Fig 1E and F).

### The 16 N-terminal amino acids determine the difference in subcellular distribution between ScGal80p and KlGal80p

It has been shown before that differences in subcellular distribution between KlGal80p and ScGal80p are an intrinsic property of the proteins and not of the host cell ScGal80p in *K. lactis* is broadly distributed (Fig 2A) and KlGal80p in *S. cerevisiae* is nuclear (Anders, 2006a; Anders et al, 2006b), but both Gal80p homologs inhibit the heterologous Gal4p variant by interaction with the conserved Gal80p binding site in the C-terminal transcription activation domain. Here we show that the N-terminal 16 amino acids are responsible for the subcellular localization of the Gal80p variants. The ScGal80 N terminus (aa 2–36) is unable to direct GFP exclusively to the nucleus (Fig 2B) whereas a chimeric protein of ScGal80p with the N terminus of KlGal80p (KlNTScGal80p) replacing the ScGal80p N terminus accumulated in the *K. lactis* nucleus as efficiently as GFP-KlGal80p itself, whereas the mutated variant KlNTScGal80-K5A/R6Ap was also detected in the cytoplasm (Fig 2D and E). This indicates (i) that ScGal80p does not have properties that prevent its nuclear accumulation and (ii) that these 16 amino acids are sufficient to cause nuclear accumulation of ScGal80p.

The N-termini of ScGal80p and KlGal80p are similar but deviate at seven positions within the first 16 amino acids (Fig 2 top). Two basic residues, which are present in KlGal80p (K8, R16) and partly constitute its NLS consensus motif $^5$KRSK$^8$, are not conserved in ScGal80p (S8, A16). To test whether the ScGal80p motif $^5$KRSS$^8$ can be converted into an NLS the serine at position 8 was mutated to lysine giving the protein variant ScGal80-S8Kp. This protein, fused to GFP, was not exclusively nuclear but GFP fluorescence showed some nuclear accumulation like full-length ScGal80p (Figs 2C and A and S3). Hence, the lysine at position 8 together with K5 and R6 contributes to nuclear transport in *K. lactis* but was not sufficient to create a strong NLS at the N terminus of ScGal80p.

### Galactose induction is impaired in the *Klgal80-K5A/R6A* mutant

To address the question whether the reduced nuclear accumulation caused by the mutation in the KlGal80-NLS had an influence on the KlGal80p inhibitor function KlGal4p controlled gene expression was examined (Fig 3). *LAC4*, one of the KlGal4p-regulated genes, encodes the *K. lactis* β-galactosidase, which allows easy monitoring of KlGal4p activity in vivo. Up to a certain limit, the intensity of the blue color correlates very well with the *LAC4* transcript level, which in turn reflects the activity status of KlGal4p (Schaffrath & Breunig,

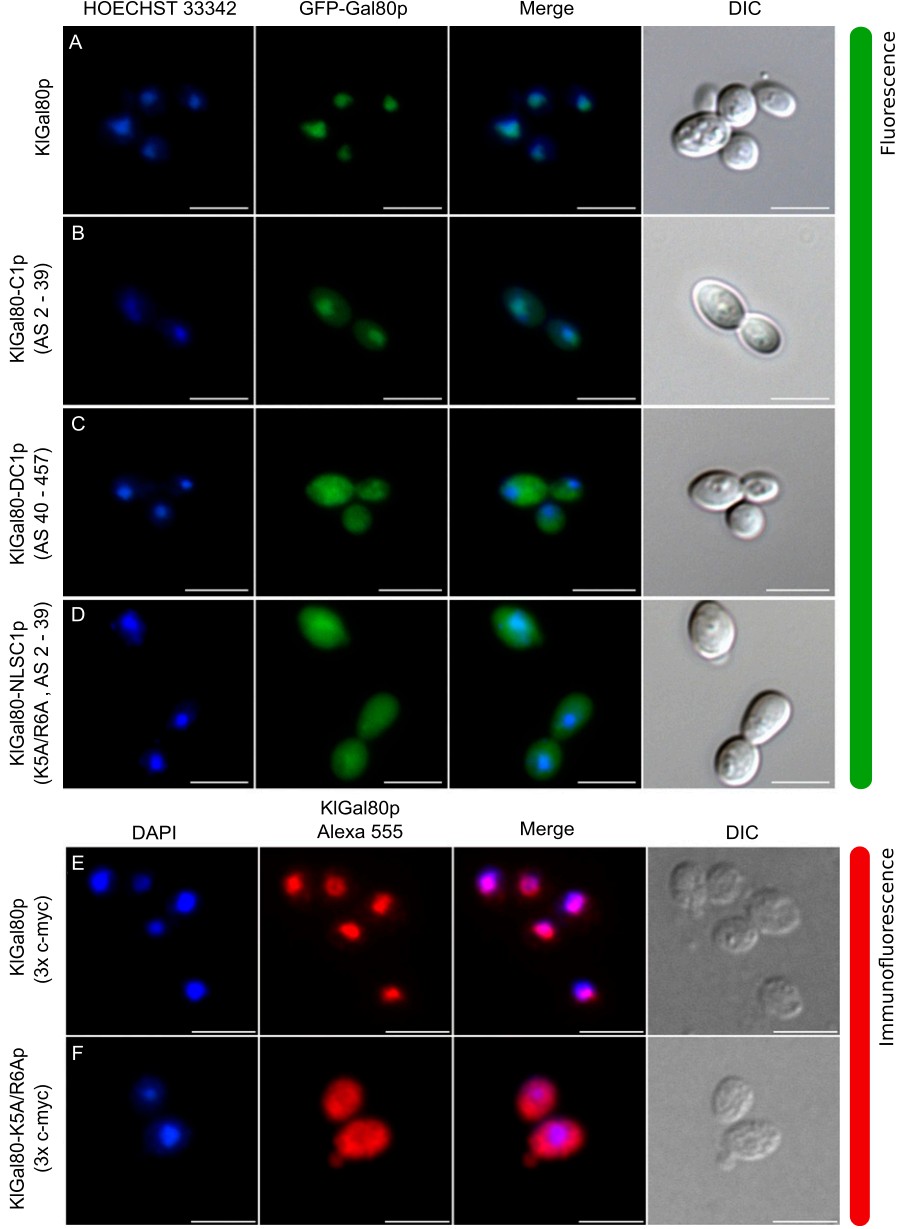

**Figure 1. Identification of an N-terminal NLS in KlGal80.**

**(A, B, C, D)** The plasmids encoding GFP fused to (A) wild-type KlGal80p (pEG80WTGFPct), (B) the KlGal80 fragment aa 2–39 (pEQRS80C1), (C) the KlGal80 fragment aa 40–457 (pEQRS80DC1), and (D) the KlGal80 fragment 2–39 with exchanges K5A/R6A (pEgal80NLS1C1), all expressed under control of the *S. cerevisiae* ADH1 promoter were transformed into the Klgal80 deletion strain JA6/D802R Cells were grown in minimal medium with 2% glucose and analyzed by fluorescence microscopy. **(E, F)** Strain JA6/G80M, in which the chromosomal *KlGAL80* gene under its own promoter was modified to express a c-myc–tagged version of wild-type KlGal80p or (F) of KlGal80K5A/R6Ap (strain JA6/G80-KR56A) were grown in 2% galactose and analyzed by immunofluorescence microscopy with a primary c-myc antibody and a secondary Alexa 555 antibody. The nucleus was stained by Hoechst 33342 or DAPI. Both channels were merged and are displayed together with differential interference contrast (DIC) view. Scale bar: 5 *µm*.

2000). *K. lactis* wild-type cells are white on glucose medium with X-Gal (because KlGal4p is inactive) and blue in the presence of an inducing sugar (lactose or galactose) because of KlGal4p activation and induction of the *LAC4* gene (Fig 3A). A *Klgal80* deletion mutant is blue on both media because, due to the absence of Gal80p, KlGal4p is constitutively active. Both Gal80p variants, KlGal80p as well as ScGal80p, inhibit gene activation by binding to DNA-bound Gal4p in the nucleus. Hence, it was expected that the mutation in the KlGal80p NLS eventually might show weaker repression than wild-type KlGal80p because of lack of nuclear accumulation. To our surprise, the opposite was the case. The *K. lactis* strain JA6/G80-KR56A containing the c-myc–tagged *Klgal80-K5A/R6A* mutant allele was white also on galactose medium, indicating a super-repressed phenotype. We can exclude that this phenotype is caused by an

elevated concentration of the KlGal80-K5A/R6A protein; the Western blot indicated no difference to wild-type KlGal80p (Fig 3B). Hence, the *Klgal80-K5A/R6A* mutant appears to be unable to respond to the presence of galactose.

β-galactosidase activity measurements over time in cells shifted from glucose to galactose medium (Fig 3C) confirmed that in the wild-type cells there is an up to 700-fold induction of β-galactosidase activity. In contrast, the *Klgal80-K5A/R6A* mutant (strain JA6/G80-KR56A) showed only a weak short-term response (13-fold increase in 2 h) and no sustained induction over a period of at least 8 h. To test the influence of the K5A/R6A mutation on ScGal80p function the K5A/R6A mutation was also introduced into the chromosomal *ScGAL80* gene of an *S. cerevisiae* strain. To have the same read-out as in *K. lactis*, this strain carried the *LAC12* and *LAC4* genes of *K. lactis*

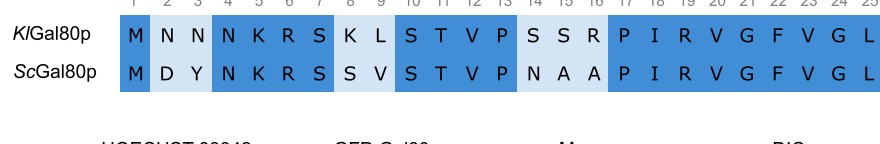

**Figure 2. Comparative analysis of the N-termini of KlGal80p and ScGal80p.**
To analyze the function of the ScGal80p N terminus plasmids encoding GFP fused ScGal80p-variants were transformed into the *K. lactis Klgal80* deletion strain JA6/D802R and localized by fluorescence microscopy. **(A, B, C, D, E)** Cells expressing the plasmids (A) pEScG80 (wild-type ScGal80p), (B) pEScG8036 (ScGal80 fragment aa 2–36), (C) pEGFPScG80-S8K (ScGal80p with amino acid exchange S8K), (D) pEGFP-KlNT-ScG80 (ScGal80p with KlGal80p N terminus aa 1–16) or (E) pEGFP-Kl56-ScG80 (ScGal80p with mutated KlGal80p N terminus containing K5A/R6A exchange) were grown in minimal medium with 2% glucose and analyzed by fluorescence microscopy. The nucleus was stained by Hoechst 33342. DIC, differential interference contrast. Scale bar: 5 µm.

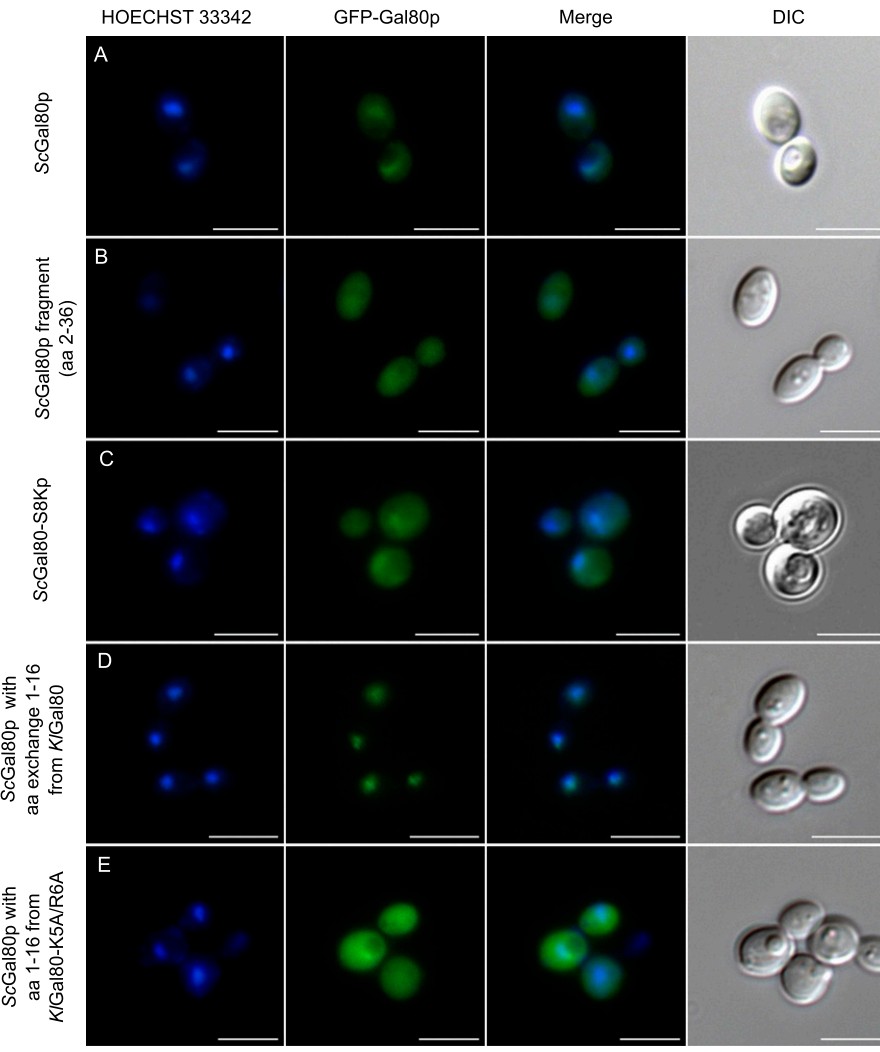

and either the *ScGAL80* wild-type or the *Scgal80-K5A/R6A* mutant allele. On X-Gal–containing plates, both strains looked identical, they were white on glucose and blue on galactose medium (Fig 3A, lower panel). Hence, in contrast to *K. lactis*, the K5A/R6A mutation in ScGal80p did not affect Gal4p activation in response to galactose.

### The super-repressed phenotype is not caused by the altered localization of KlGal80-K5A/R6Ap

Next, we asked if in *K. lactis* the super-repressed phenotype of the *Klgal80-K5A/R6A* mutant is caused by the destruction of the NLS and thus by the resulting changes in intracellular localization of the mutated Gal80 protein. Therefore, we tried to redirect KlGal80-K5A/R6Ap to the nucleus by fusing the strong SV40-NLS (MGAPPKKKRKVA)

to wild-type KlGal80p and to the mutated variant KlGal80-K5A/R6Ap, respectively. Immunofluorescence studies with *K. lactis* cells showed that the SV40-NLS did not interfere with the KlGal80-NLS, as wild-type KlGal80p with and without the SV40-NLS were nuclear (Fig S2A and C). Importantly, nuclear accumulation of mutant KlGal80-K5A/R6Ap could indeed be restored by the SV40-NLS (Fig S2D).

We tested the ability of the KlGal80p variants to regulate KlGal4p controlled gene expression by measuring β-galactosidase activity again after shifting from glucose to galactose (Fig 3C). In *K. lactis* the fusion of KlGal80p with the heterologous SV40-NLS reduced galactose induction, but clearly did not prevent the response to galactose. The strains expressing the mutated KlGal80-K5A/R6Ap variant with or without the SV40-NLS, JA6/G80-SVKR and JA6/G80-KR56A, respectively, only showed a weak, short-term response to induction. Hence, restoring the nuclear localization of the defective

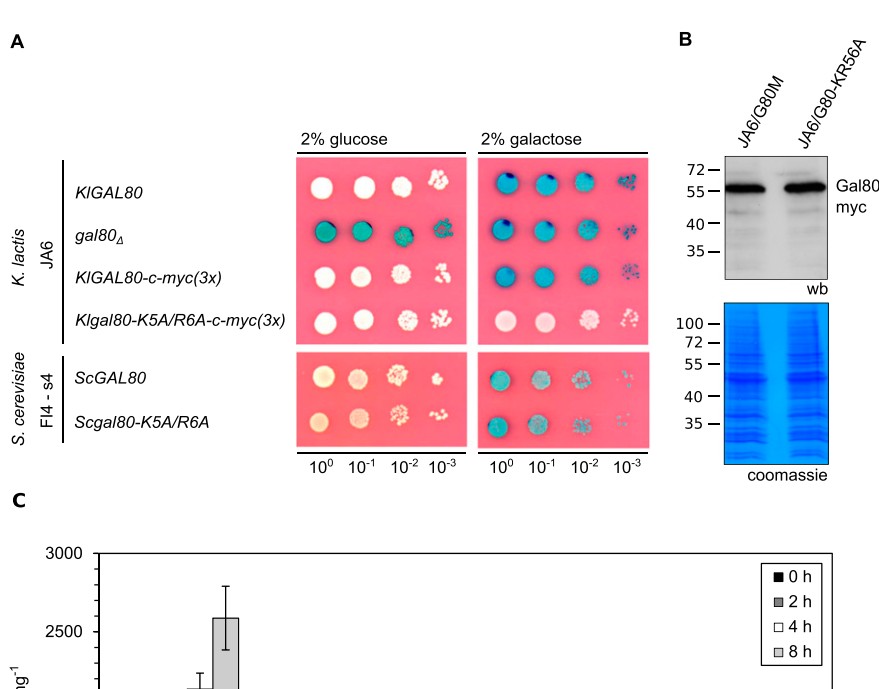

**Figure 3. Influence of Gal80p localization on Gal4p-controlled *β*-galactosidase gene expression.**
**(A)** The *K. lactis* wild-type strain JA6 and derived mutated strains JA6/D802 (*Klgal80Δ*), JA6/G80M (c-myc–tagged *KlGAL80*) and JA6/G80-KR56A (c-myc–tagged *Klgal80-K5A/R6A*) and the *S. cerevisiae* strain FI4 s4 containing the *K. lactis LAC4-LAC12* genes and expressing wild-type ScGal80p or ScGal80-K5A/R6Ap were analyzed in an X-Gal plate assay. The plates were incubated for 3 d at 30°C. **(B)** Comparison of KlGal80p and KlGal80-K5A/R6A protein levels by Western blotting. Strains JA6/G80M expressing wild-type KlGal80p and JA6/G80-KR56A expressing *Klgal80-K5A/R6Ap* under their own promoters and triple myc-tagged were grown in glucose overnight. Protein extracts were prepared and analyzed by polyacrylamid gel electrophoresis and Western blotting with c-myc antibodies (Top) or Coomassie staining. **(C)** Strains expressing triple-myc-tagged wild-type Gal80p (JA6/G80M), Gal80-K5A/R6Ap (JA6/G80-KR56A), or the corresponding variants fused to SV40-NLS (JA6/G80-SV40) or (JA6/G80-SVKR) were grown in glucose and shifted to galactose-containing medium. Beta-galactosidase activity was measured in crude extracts of samples taken at the indicated time points (hours) after the galactose shift. Mean values and standard variations based on two independent measurements each measuring three different dilutions with four technical replicates (12 replicates in total) are plotted.

KlGal80-K5A/R6A protein did not restore inducibility, indicating that the super-repressed mutant phenotype is not caused by the influence of the NLS-mutation on nuclear transport of KlGal80p (Fig S3).

## KlGal80p controls the intracellular distribution of KlGal1p

Both in the presence and in the absence of KlGal80p, a GFP-KlGal1p fusion protein can be detected in the entire cell (Anders et al, 2006b) (Fig 4A and B). This changes with galactose induction or constitutive overexpression of the *KlGAL80* gene: both conditions lead to nuclear accumulation of GFP-KlGal1p (Fig 4C) (Anders et al, 2006b). Apparently, the KlGal1p subcellular distribution is influenced by the concentration of KlGal80p.

Here we have shown that no nuclear accumulation was observed (Fig 4D) when the mutated *Klgal80-K5A/R6A* allele was introduced in multicopy into a *Klgal80 Klgal1* mutant constitutively expressing the GFP-KlGal1p fusion protein. Because KlGal80-K5A/R6Ap itself does not accumulate in the nucleus (Fig 1F), we tested whether the SV40-NLS fusion, which partially restored nuclear accumulation of the KlGal80p-K5A/R6Ap variant, was able to promote nuclear accumulation of GFP-KlGal1p when overexpressed. This could indeed be shown (Fig 4E). We also assayed a *K. lactis* mutant, *Klgal80-s2*, that is KlGal1p binding deficient because of an E367K substitution (Zenke et al, 1996). The KlGal80-E367K protein has an intact N-terminal NLS and is nuclear. In contrast to nuclear KlGal80-K5A/R6A-SV40p, it did not accumulate GFP-KlGal1p in the nucleus (Fig 4F). We conclude that accumulation of KlGal1p requires (i) effective nuclear transport of KlGal80p, which can be achieved by the intact N-terminal NLS, or by a heterologous NLS, and (ii) the ability of KlGal1p and KlGal80p to form a complex. In summary, we propose that KlGal1p enters the nucleus piggy-backing on KlGal80p.

## The non-inducible phenotype of the *Klgal80-K5A/R6A* mutant is suppressed by overexpression of KlGal1p

To further elucidate why the NLS mutant is non-inducible, we analyzed the influence of *KlGAL80* and *KlGAL1* gene dosage on KlGal4p activity using the X-Gal plate assay (Fig 5). The *K. lactis* strain JA6/G80M (*KlGAL80-(c-myc)₃*) (Fig 5A) and the otherwise isogenic strain JA6/G80-KR56A (*Klgal80-K5A/R6A-(c-myc)₃*) (Fig 5B) were

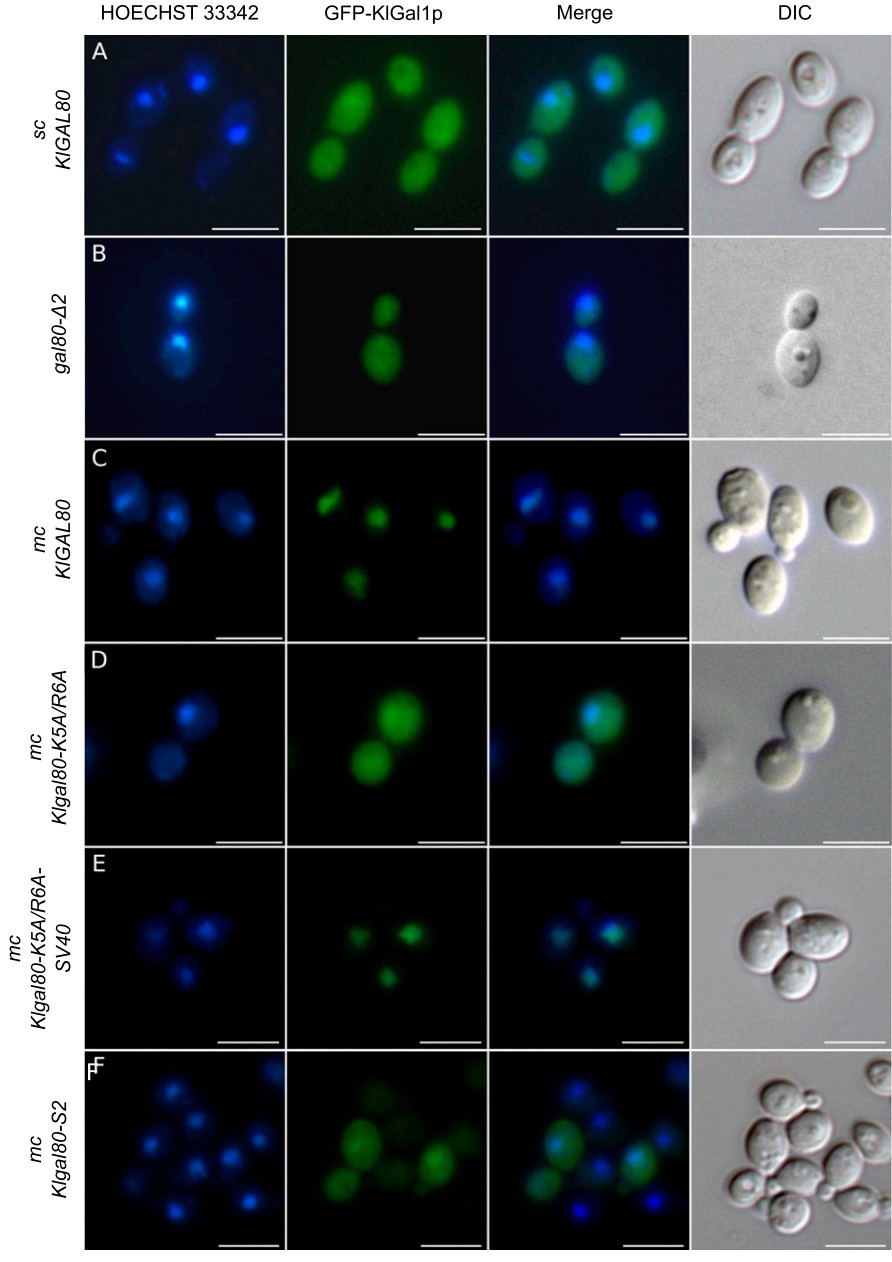

| HOECHST 33342 | GFP-KlGal1p | Merge | DIC |
|---|---|---|---|

*sc KlGAL80* **A**

*gal80-Δ2* **B**

*mc KlGAL80* **C**

*mc Klgal80-K5A/R6A* **D**

*mc Klgal80-K5A/R6A-SV40* **E**

*mc Klgal80-S2* **F**

**Figure 4. Influence of *Klgal80* mutations on KlGal1p subcellular localization.**
The KlGal1-GFP fusion protein was localized by fluorescence microscopy in cells expressing KlGal80p variants. **(A, B, C, D, E, F)** The KlGAL1-GFP protein was constitutively expressed from a centromeric plasmid pCGFPAG1 (A, B) or pCGFPAG1-ura3Δ (C, D, E, F), the KlGal80p variants from a multicopy plasmid, both genes under control of the constitutive *ScADH1* promoter in *K. lactis* strain JA6/D1R (A) deleted for the chromosomal *KlGAL1* gene, or JA6/D1D802R deleted for *KlGAL1* and *KlGAL80* (A, C, D, E, F). KlGal80p variants were wild-type (expressed from chromosomal *KlGAL80* gene) (A); wild-type from multicopy plasmid pEAG80 (C); Gal80-K5A/R6Ap (from pEAG80-KR56A) (D); Gal80-K5A/R6Ap with N-terminal SV40-NLS-fusion (from pEAG80-KR56A-SV40) (E) or KlGal80$^{S2}$p (E367K) from pEAG80S2 (F). The nucleus was stained with Hoechst 33342, KlGal1p was detected by its GFP-tag. sc, single copy; mc, multicopy. Merge: GFP and Hoechst channels were superimposed. DIC, differential interference contrast. Scale bar: 5 $\mu$m.

transformed with single copy (sc) or multicopy (mc) plasmids containing the *KlGAL80* or the *KlGAL1* gene under control of the strong, constitutive *ADHI* promoter from *S. cerevisiae*. In the *KlGAL80* wild-type background expressing a (c-myc)$_3$-tagged version of KlGal80p under its own promoter (Fig 5A, empty vector) the color shift from white on 2% glucose to dark blue on galactose indicates the release of KlGal4p from KlGal80p inhibition resulting in galactose induction of the *LAC*4 gene. The unexpected blue color on 0.2% glucose in the empty vector control was partially suppressed by one additional gene copy (sc GAL80, on a centromeric vector under *ScADH1* promoter control) and fully suppressed by multicopy *KlGAL80* (mc GAL80, episomal vector) indicating incomplete KlGal4p inhibition by KlGal80p under these growth conditions. In the JA6/G80-KR56A (*Klgal80-K5A/R6A-*(*c-myc*)$_3$)

mutant strain (Fig 5B), the intensity of the blue color correlated with growth on low galactose medium. The poor growth of the empty vector control confirmed the negative influence of the NLS mutation on KlGal4p activation, which impairs induction of the galactose metabolic genes. Single-copy *KlGAL80* (sc GAL80) weakly improved galactose induction and growth on low galactose indicating that the mutant phenotype is recessive to the wild type. With multicopy *KlGAL80* (mc GAL80) the known negative influence of KlGal80p overexpression on KlGal4p activation (Zenke et al, 1993) becomes apparent.

Most importantly, in the *Klgal80-K5A/R6A* mutant background the non-inducible and the slow growth phenotypes were effectively suppressed by multicopy *GAL1* (mc GAL1). *KlGAL1* on a centromeric vector (sc GAL1) resembled the empty vector control and only had a weak effect.

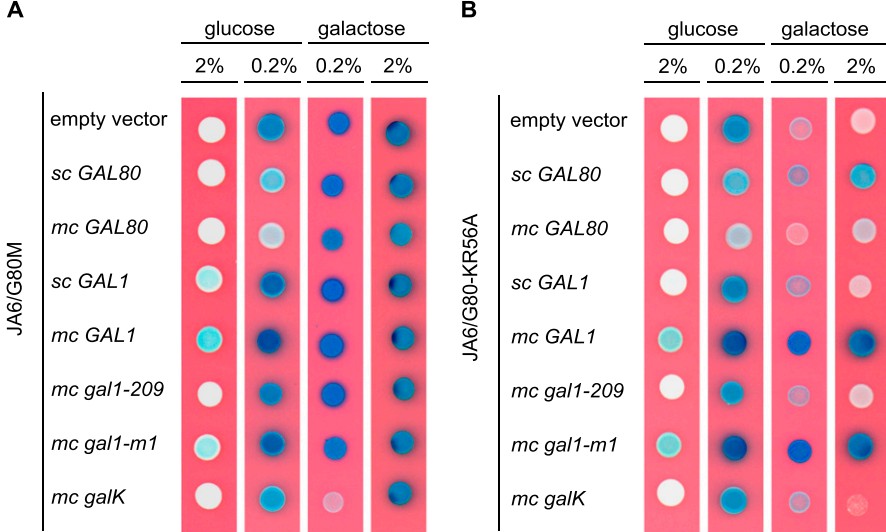

**Figure 5. Influence of the *KlGAL1* gene on Gal4p controlled *β*-galactosidase gene expression in *KlGAL80* wild-type and *Klgal80-K5A/R6A* mutant.** **(A, B)** The *K. lactis* strains (A) JA6/G80M (c-myc–tagged *KlGAL80*) and (B) JA6/G80-KR56A (c-myc–tagged *Klgal80-K5A/R6A*) were transformed with the multicopy (pE) or centromeric (pC) plasmids pE1 (empty vector), constitutively expressing Gal80p under *ADH1* promoter control, pC80 (*sc KlGAL80*), pEAG80 (*mc KlGAL80*), pCAG1 (*sc KlGAL1*), pEAG1 (*mc KlGAL1*), pEAG1-209 (*mc Klgal1-209*, N261Y), pEAG1-m1 (*mc Klgal1-m1*, E254K), and pEAGK1 (*mc galK* galactokinase from *E. coli*). The resulting colonies were analyzed in an X-Gal plate assay on YNB plates without uracil. The plates were incubated for 3 d at 30°C. sc, single copy; mc, multicopy.

We have also tested two *Klgal1* mutants for their ability to suppress the non-inducible phenotype of the *Klgal80-K5A/R6A* strain. The *Klgal1-m1*(E254K) mutant has been characterized as catalytically active but regulatory deficient whereas, conversely, the *Klgal1-209* (N261Y) mutant has been shown to be catalytically inactive but regulatory proficient. Because the KlGal1-m1(E254K) protein had reduced affinity for KlGal80p in vitro and was a poor substitute for KlGal1p in a *KlGAL80* wild-type background (Vollenbroich et al, 1999; Menezes et al, 2003), we expected that it would not be able to suppress the non-inducible phenotype of the *Klgal80-K5A/R6A* strain but did suppress like wild-type KlGal1p (Fig 5B). In contrast, the *Klgal1-209*(N261Y) mutant was unable to suppress. The KlGal1-209 (N261Y) protein has no galactokinase activity but displayed galactose and ATP dependent binding to KlGal80p and galactose-inducible activation of KlGal4p in combination with wild-type KlGAL80p (Fig 5A) (Zenke et al, 1996, 1999; Menezes et al, 2003). However, in the *Klgal80-NLS* mutant, the overexpression of the mutated KlGal1p variants gave phenotypes exactly opposite to our expectations (Fig 5B). Because these results raised the question if galactokinase activity was more important for suppression than the ability to bind KlGal80p, we also tested whether the galactokinase gene from *Escherichia coli* (mc *GalK*) in multicopy could suppress the *Klgal80-K5A/R6A* mutant phenotype but this was not the case.

### The K5A/R6A mutation in KlGal80p affects KlGal1p interaction

We expressed and purified N-terminally and internally His6-tagged as well as untagged versions of wild-type KlGal80p (N-terminal His6-tag: NHKlGal80p, internal His6-tag: IHKlGal80p, untagged: KlGal80p), the K5A/R6A variant (NHKlGal80-K5A/R6Ap) and NHKlGal1p in bacteria to compare the affinities of wild-type and mutated KlGal80p for KlGal1p in

vitro. As described before, the interaction of KlGal80p with KlGal1p inhibits the galactokinase activity of this enzyme and can be used to quantify the interaction between the KlGal80p variants and KlGal1p proteins (Anders et al, 2006b). Titration of increasing amounts of KlGal80p into a galactokinase assay resulted in a first order inhibition curve which enabled determination of dissociation constants (Fig 6A and B). Wild-type KlGal80p tightly binds KlGal1p ($K_D$ ~80 nM), whereas the NHKlGal80-K5A/R6Ap indicates almost 40-fold weaker binding ($K_D$ ~3.1 $\mu$M). To test whether the N terminus of KlGal80p alone can bind KlGal1p, we used a synthetic peptide corresponding to amino acid 1–17 of KlGal80p. The presence or absence of that peptide did not show any difference in KlGal80p-mediated galactokinase inhibition in vitro (Table 1), suggesting that the N terminus is essential, but not sufficient for the interaction with KlGal1p.

A lower affinity of the NHKlGal80-K5A/R6Ap to KlGal1p than wild-type was confirmed in a GST-pull-down assay where Gst-KlGal1p was used as a bait for purified untagged KlGal80p and KlGal80-K5A/R6Ap (Fig 6C). Whereas wild-type KlGal80p could be detected in the bound fraction, this was not the case for KlGal80-K5A/R6Ap. There was no difference whether KlGal80p and KlGal80-K5A/R6Ap were untagged or had a His6-tag (data not shown). Because the mutated KlGal80-K5A/R6Ap variant had the same thermostability as wild-type KlGal80p (Fig S4), we conclude that the KR>AA exchange is responsible for the reduced affinity of KlGal80-K5A/R6Ap for Gst-KlGal1p in vitro. In a similar pull-down experiment with Gst-Gal4p, KlGal80p could be found in the bound fraction but there was no difference between wild-type KlGal80p and KlGal80-K5A/R6Ap (data not shown).

Because KlGal4p was known to have a much higher affinity for KlGal80p than KlGal1p (Anders, 2006a), we tried, in a reverse approach, to dissociate KlGal80p from a preformed KlGal1p complex. KlGal1p was pre-incubated with KlGal80p wild-type and mutant

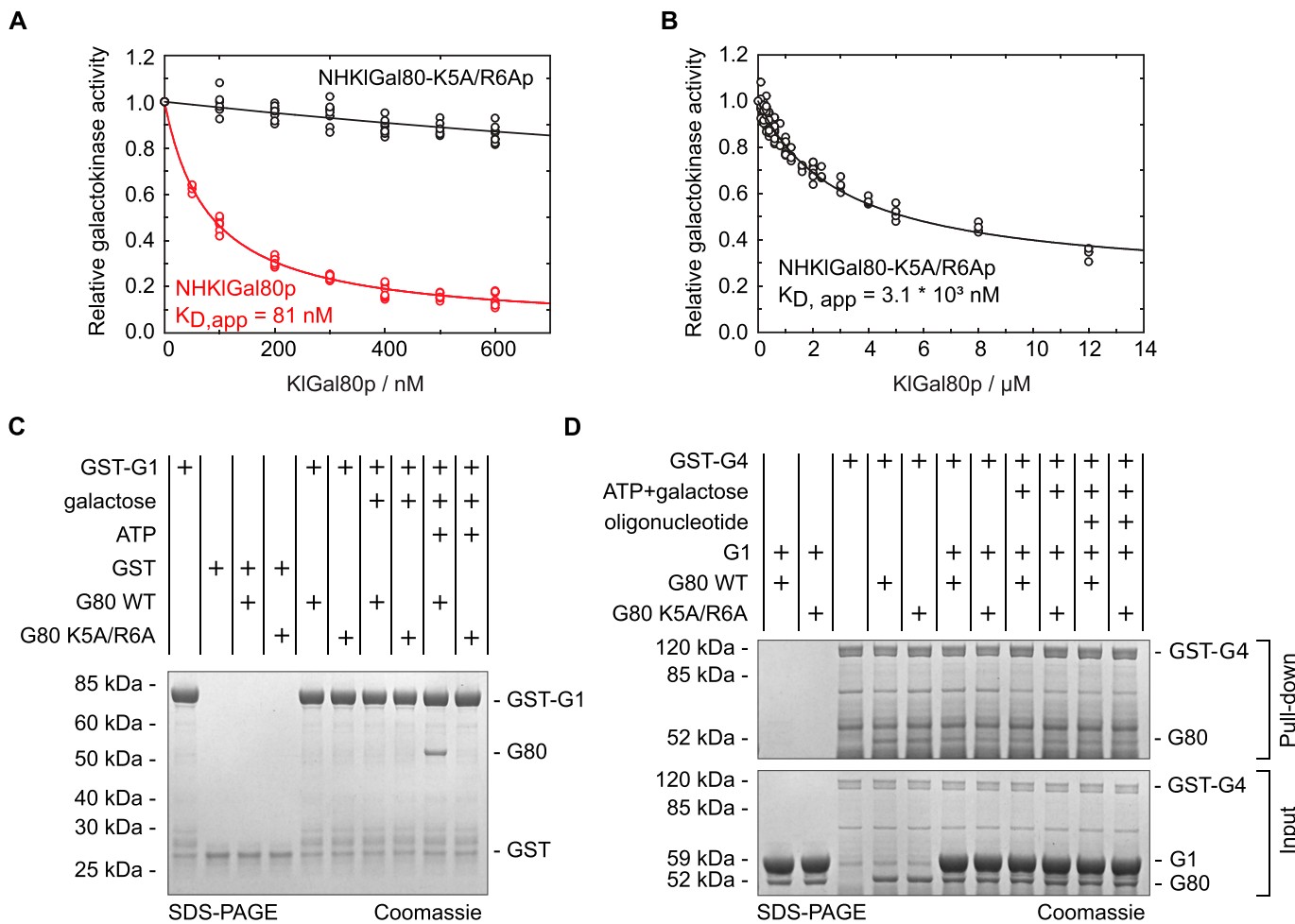

**Figure 6. Influence of the KlGal1-NLS mutation on *Kl*Gal1p interaction.**
**(A, B)** Recombinant proteins produced in *E. coli* (KlGal80p with N-terminal His₆-tag, NHKlGal80p, NHKlGal80-K5A/R6Ap, and Gst-KlGa1p) were purified to determine the affinity of KlGal80p and KlGal80K5A/R6Ap for KlGal1p in vitro by making use of the inhibitory effect of KlGal80p on KlGal1p's galactokinase activity. Enzymatic activity was determined as described previously (Anders et al, 2006b). Data points result from two independent measurements with four replicates each. K$_{D, app}$ values were calculated from the decline in KlGal1p activity with increasing KlGal80p (red, NHKlGal80p) or KlGal80-K5A/R6Ap (black, NHKlGal80-K5A/R6Ap) concentrations and were. K$_{D, app}$ = 81 nM for KlGal80p wild-type and K$_{D, app}$ = 3.1 × 10³ nM for KlGal80-K5A/R6Ap. **(C)** The interaction of purified wild-type (IHGal80p) and mutant (IHGal80-K5A/R6Ap) KlGal80p with GST-tagged KlGal1p was analyzed by a GST-Pull-Down-Assay. 1,500 µg raw extract containing GST-Gal1p was immobilized on a GST-column for 1 h 30 min and then washed twice. 20 µg of purified Gal80p was added and incubated for 1 h 30 min. The column was washed four times and GST-Gal1p was eluted with buffer containing 40 mM GSH. 165 µg of raw extract from JA6/D1D802R transformed with pGSTGal1, 2 µg of purified KlGal80p as well as 10 µl of flow through, washing and elution fractions were loaded on an SDS-Gel and stained with Coomassie solution after PAGE. FT, flow through; WF, wash fraction. **(D)** Pull-down experiment with Gst-KlGal4p (full length KlGal4p) comparing KlGal80p wild-type and NLS mutant protein. Before the binding step untagged KlGal80p and KlGal80-K5A/R6A were mixed with a threefold molar excess of untagged KlGal1p to preoccupy Gal80p with Gal1p in the presence or absence of the Gal1p ligands galactose (2% wt/vol) and ATP (5 mM) as indicated. After a preincubation step with the two proteins (30 min, at 30°C), Gst-Gal4p was added and samples were taken for loading control (shown in Fig S5) Gst-Gal4p was allowed to bind to glutathione-beads (GSH-Sepharose) at 4°C for 2 h, beads were collected by centrifugation at 500*g*, bound material was denatured and analyzed on 4–12% gradient gels (Invitrogen) stained with Coomassie for imaging.

variants in the presence of galactose and ATP to allow for formation of the KlGal80–KlGal1 complex before the Gst–KlGal4p fusion protein was added for pull-down. Both KlGal80p variants were detected in the Gst–Gal4p pull down fraction and the intensity of the bands representing one or the other KlGal80p variant was neither affected by the presence or absence of galactose and ATP nor by the presence or absence of KlGal1p in the assay (Fig 6D). Hence, the amount of KlGal80p in the bound fraction was only determined by the Gst-KlGal4p input (see Fig S5 for input control). We conclude that in this three-component in vitro binding assay KlGal1p cannot effectively compete with Gst-KlGal4p. Presumably,

the fact that free KlGal1p is monomeric in solution (Barnard & Timson, 2011) whereas KlGal4 and KlGal80p can form a tetrameric (KlGal4)₂ (KlGal80)₂ complex (Melcher & Xu, 2001) contributes to the difference in affinity between the two Gal80 complexes.

Taken together these findings confirm that the *Klgal80-K5A/R6A* mutation reduces the affinity to KlGal1p and decreases the KlGal80p-mediated inhibition of galactokinase activity. The latter may be an indirect consequence of the lower affinity or a direct consequence of the mutation or both. In any case, the invariable amount of KlGal80p in all samples suggests that in this vitro assay the concentration of Gst-KlGal4p in the input (Fig S5) determines how

**Table 1. Influence of a 17-aa peptide representing the KlGal80 N terminus (KlN17G80) on galactokinase inhibition by KlGal80p or KlGal80-K5A/R6Ap.**

| KlGal80p variant/peptide | Relative galactokinase activity (in %)[a] |
|---|---|
| KlGal80p | 20.27 ± 1.03 |
| KlGal80-K5A/R6Ap | 93.63 ± 4.72 |
| KlN17G80 peptide | 99.43 ± 4.92 |
| KlGal80p and KlN17G80 peptide | 18.77 ± 2.05 |

[a]The measured kinase activity was compared with the activity in the absence of KlGal80p or its peptide giving the relative kinase activity. The values result from two independent measurements with four technical replicates each.

much KlGal80p is in the bound fraction, whereas the influence of KlGal1p can be neglected. Phenotypic differences between KlGal80p and KlGal80-K5A/R6Ap may then result from the difference in stability or dynamics of the KlGal1p-KlGal80p complexes.

### The N terminus of KlGal80p can bind near the active site of KlGal1p

So far sequence conservation, point mutations that affect galactose induction, and the structure of the ScGal80–ScGal3 complex had shown that direct contacts between Gal3p and Gal80p reside mostly in the C-terminal half of both proteins (Vollenbroich et al, 1999; Menezes et al, 2003; Lavy et al, 2012). An involvement of the extreme N terminus has not been reported before. Therefore, we decided to analyze the preferential orientations of the KlGal80p N terminus on the KlGal1p–KlGal80 complex in silico. Based on the structure of the ScGal3-ScGal80 (PDB ID 3V2U) and its surface charge, the KlGal1–KlGal80 complex was modelled (Fig S6), and five potential binding sites for the KlGal80 (aa 1–16) peptide were defined (Fig S6). Subsequently, these potential binding sites were probed for interaction with the KlGal80 1-16 peptide using a FlexPepDock platform.

The five docking sites with the highest likelihood of binding are shown for the KlGal80 1-16 peptide with the wild-type and the mutated sequence (Fig S6). In the energetically most favorable conformation, the N terminus reaches the bound ATP molecule (Fig 7A and B). In this model, the interaction of the KlGal80(1-16) peptide with KlGal1p is mostly dependent on N3, K5, and R6. When docking to the KlGal1–KlGal80 complex was performed with the mutated peptide sequence KlGal80(1-16)-K5A/R6A, the active site of KlGal1p was occupied with a lower probability (Fig S6). The predictions are in line with the experimental galactokinase inhibition data (Fig 6A and B), which confirmed the decreased affinity of KlGal80-K5A/R6Ap to KlGal1p revealed by the Gst-KlGal1 pull-down experiment (Fig 6C). We propose that, in addition to an extended Gal1p-Gal80p interface involving mainly the second half of KlGal80p, there is a secondary contact site between the KlGal80p N terminus and KlGal1p. Our genetic and biochemical data support the view that the N terminus of KlGal80p interacts with the catalytic center of KlGal1p and contributes to KlGal4p activation. This is the first report for a positive role of the Gal4-negative regulator Gal80p.

## Discussion

The *Saccharomyces* and *Kluyveromyces* genomes have been shaped by more than 100 million years of divergent evolution. In that time, two genetic events occurred that undoubtedly had massive impact on the *GAL* regulon. First, the whole genome duplication in the *Saccharomyces* lineage was accompanied by rearrangements, deletions and duplications of chromosome segments (Lynch & Conery, 2000; Kellis et al, 2004), and second, the loss of the *LAC* genes disabled lactose metabolism, which occurred in *S. cerevisiae* and independently in many other ascomycetes. The structures and functions of the regulatory proteins involved in the *GAL* switch,

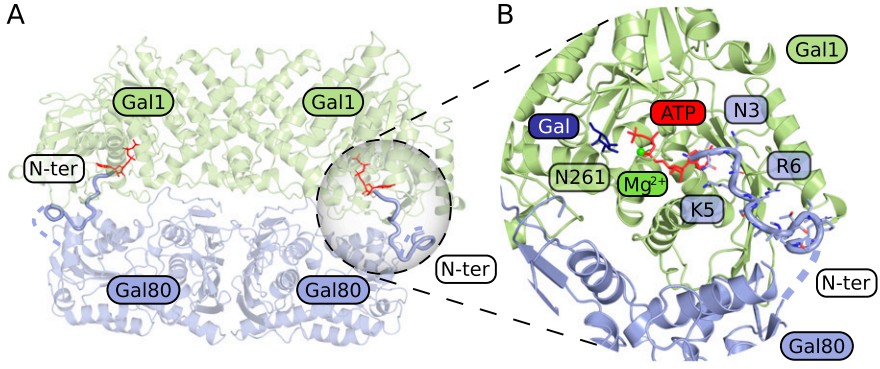

**Figure 7. Structural model of KlGal80p–KlGal1p interaction and the role of the KlGal80 N terminus.**
**(A)** Structural overview of a homology-based model of KlGal1 (based on ScGal3 in 3V2U) in complex with KlGal80 (PDB ID 3E1K). Thick cartoon (N terminus, blue) represents the best docking result of KlGal80 1-16 to the KlGal1 molecule, ATP is shown as a stick model (red). **(B)** A close-up of KlGal1p-active site in an open conformation. Galactose (dark blue), magnesium (bright green), and ATP (red) were positioned according to their position in ScGal3 (PDB ID 3V2U). Docked KlGal80 1-16 peptide clashes with ATP. Residues of potential importance for peptide–protein interactions are highlighted.

namely, Gal4p, Gal80p, and Gal1/Gal3p are very similar in *S. cerevisiae* and *K. lactis*, but there is also evidence for divergent evolution at the protein level. Here, we report data supporting the view that in detail the molecular mechanism of Gal4p activation is not identical in the two yeast species presumably because of the bifunctional KlGal1 protein.

We have asked how the reported differences in intracellular distribution between KlGal80p and ScGal80p (Anders et al, 2006b) are achieved and maintained. We were able to show that the very N terminus of KlGal80p contains an autonomous NLS able to direct a GFP-fusion protein to the nucleus. A class 2 NLS consensus [5]K-K/R-X-K/R[8] is a necessary functional element of the KlGal80-NLS. It was sufficient to swap the 16 N-terminal amino acids of ScGal80p and KlGal80p to exchange their intracellular distribution giving ScGal80p an exclusive nuclear KlGal80p-like localization and vice versa (Anders et al, 2006b). Unexpectedly, mutating the basic amino acids K5 and R6 in the NLS not only changed the localization of KlGal80p from exclusively nuclear to nucleocytoplasmic but also impaired KlGal4p activation. Because the super-repressed phenotype was not affected by redirecting KlGal80-K5A/R6Ap to the nucleus, we concluded that impaired activation of KlGal4p in the mutant is not due to impaired nuclear import of KlGal80-K5A/R6Ap but to a so-far unknown property of the N terminus of KlGal80p.

Here, we show that the N-terminal region including the NLS consensus sequence is not only important for nuclear transport of KlGal80p but also for nuclear transport of KlGal1p and for the KlGal80p-mediated inhibition of KlGal1p's galactokinase activity. These functions are overlapping and all of them are affected by the introduced NLS-mutation. The negative influence of this mutation on Gal4p activated gene expression identified the N terminus of *K. lactis* Gal80p as an important new element in KlGal4p activation.

Support for a co-transport of KlGal1p with KlGal80p comes from dominant mutations in the *GAL80* gene that give a super-repressed phenotype. Three super-repressed *ScGAL80[S]* mutants have been described long ago (Nogi et al, 1977). Among those the *K. lactis* equivalent of the *ScGAL80-s2* (E351K) mutation, *Klgal80-E367K*, also confers super-repression in *K. lactis* (Zenke et al, 1993). The E351 residue is located at the major interface of the ScGal80–Gal3 complex and makes direct contact with Gal3p, which is disrupted by the E351K mutation (Lavy et al, 2012). The resulting super-repression is therefore explained by the inability of KlGal1p or ScGal3p to form a stable complex with the mutated Gal80 protein. The mutation is dominant because the Gal80-s2 protein remains bound to Gal4p and blocks the Gal4 activation function even in the presence of wild-type Gal80p (Nogi et al, 1977; Menezes et al, 2003). In contrast, the super-repressing/non-inducible phenotype caused by the *Klgal80-NLS* mutation is recessive, indicating that the wild-type KlGal80 protein can replace the NLS-deficient variant in the KlGal1–KlGal80 complex. The finding that the KlGal80-K5A/R6Ap but not the KlGal80-E367Kp is able to accumulate KlGal1p in the nucleus supports the view that the KlGal80–KlGal1 complex formation is more severely affected by the KlGal80-E367K replacement than by the NLS mutation.

We believe that the KlGal80p-mediated nuclear KlGal1p accumulation is a good indicator for in vivo interaction between KlGal1p and KlGal80p. We take the ability of the KlGal80-K5A/R6A protein to accumulate KlGal1p in the nucleus as support for an interaction between KlGal80p and KlGal1p that is retained when the NLS is mutated. Most likely, their nuclear co-transport is possible because

the two proteins are held together in vivo by the extended major interface, which is not affected by the mutation. We propose that the K5A/R6A mutation affects a separate, secondary interaction site, which seems to be required for activation of KlGal4p as indicated by the super-repressed phenotype of the NLS mutant. This would be the first indication for a positive role of Gal80p in Gal4p regulation.

Our in silico docking experiments show that, in principle, the structurally ill-defined N terminus of KlGal80p can reach out to the catalytic center of the galactokinase enzyme. A KlGal80p N-terminal peptide (1-16) virtually linked to the modelled KlGal1–KlGal80 complex found its preferred docking site oriented towards the bound ATP (Fig 7A and B) whereas with the mutated peptide this preference was not seen. Biochemical experiments (Fig 6), the results of which were not considered in the modelling, are compatible with our in silico results and support the modelled interaction between the wild-type KlGal80 N terminus and the catalytic center of KlGal1p. We hypothesize that the amino acid exchange KR>AA reduces KlGal80p-mediated enzyme inhibition by affecting a direct interaction of the N terminus with the KlGal1p catalytic center.

We consider two models, to explain how the wild-type N terminus of KlGal80p could support KlGal4p activation, a structural one and a kinetic one, which are not mutual exclusive. The structural model implies that a transition of KlGal80p from the KlGal4 complex to the KlGal1 complex, as proposed for ScGal80p (Jiang et al, 2009; Abramczyk et al, 2012), requires a conformational change in the KlGal1-KlGal80 complex. Such a change could be triggered by the interaction of the KlGal80 N terminus with KlGal1p, possibly accompanied by the conformational change described for galactokinase enzymes including ScGal1p and KlGal1p (Holden et al, 2004; Thoden et al, 2005). The change in Gal1p is induced by binding of the galactokinase substrates galactose and ATP and associated with a ligand-induced closed-open-closed conformation characteristic for the catalytic cycle. In the crystal structure, ScGal80p was found to bind Gal3p in the closed conformation, which has a higher affinity for ScGal80p (Lavy et al, 2012). However, whether the higher affinity suffices to promote dissociation of Gal80p from its high-affinity binding site on Gal4p in vivo is highly disputed.

Trying to reconstitute the postulated transition in vitro we have purified the regulatory proteins including full-length KlGal4p and the mutated variants of KlGal80p in His$_6$-tagged and untagged form recombinantly expressed in *E. coli*. We could reconstitute KlGal4 and KlGal1 complexes with wild-type and mutated KlGal80p variants and demonstrated the reduced affinity of the mutated KlGal80p for KlGal1p (Fig 6B and C). However, we were unable to show in vitro that KlGal1p or the KlGal1–KlGal80 complex can dissociate KlGal80p from KlGal4p. KlGal1p, even when present in excess and allowed to form a complex with KlGal80p before Gst-KlGal4 was added, did not have any measurable influence indicating that in vitro the affinity of KlGal80p for KlGal1p is much lower than for KlGal4p. We have no indication for a difference between wild-type KlGal80p and the NLS mutant as far as Gal4p binding is concerned. If a difference exists it is too small to be detected in our assay. Therefore, we assume that influence of the NLS mutation on KlGal4p activation is primarily caused by the difference in affinity for KlGal1p. This view is supported by our genetic data. Notably, the influence of elevated *KlGAL80* gene dosage, as measured by *LAC4* gene expression (Fig 5B), is very similar in the *KlGAL80* wild-type and the *Klgal80-K5A/*

*R6A* mutant in low glucose medium, whereas in galactose, there is a marked difference and a strong allele-specific influence of the *KlGAL1* gene dosage. The well-known Gal4p inhibitor function of KlGal80p, which blocks the transcription activation function of KlGal4p under non-inducing conditions is hardly affected by the NLS mutation. This mutation primarily inhibits the release of KlGal4p from KlGal80p-mediated inhibition, which requires KlGal80-KlGal1 interaction. Because the induction deficiency associated with the *Klgal80-K5A/R6A* mutation can be suppressed by overexpression of KlGal1p, we conclude that a high level of KlGal1p does not only lead to partial KlGal4p activation under non-inducing conditions it can also compensate the functional inactivation of the KlGal80 N terminus by the NLS mutation (Fig 5A and B).

It is well documented that in *S. cerevisiae* and in *K. lactis*, gene regulation by Gal4p is strongly influenced by the concentration and the affinity of Gal80 and Gal1/3 (Bhat & Hopper, 1992; Vollenbroich et al, 1999b; Verma et al, 2003; Bhat & Venkatesh, 2005; Anders et al, 2006b). In our kinetic model, we propose that the interaction of the wild-type KlGal80 N terminus with the catalytic center of KlGal1p has a negative effect on the dynamics of the KlGal1–KlGal80 complex, in other words, the wild-type N terminus has a stabilizing function for the complex, which is lost by the NLS mutation. Assuming no or little influence of the NLS mutation on the KlGal80-KlGal4 interaction, the phenotypic difference between the KlGal80 wild-type protein and the NLS-mutated variant would be caused by the difference in affinity for KlGal1p. The kinetic model explains the non-inducible phenotype of the NLS mutant by a higher dissociation rate of the KlGal80-K5A/R6A–KlGal1 complex, which can provide elevated levels of KlGal80p able to amplify KlGal4p inhibition under inducing conditions. A high association rate and a low dissociation rate of the Gal80–Gal4 complex could lead to sequestration of Gal80p to Gal4p further supporting super-repression.

The fact that the non-inducible phenotype of the NLS mutant can be suppressed by overexpression of KlGal1p indicates that the lower affinity of the KlGal80-K5A/R6A protein for KlGal1p can be compensated by the higher concentrations of KlGal1p. The kinetic model predicts that the dynamics of the KlGal80-KlGal1 complex is very important for the high copy suppression by KlGal1p. As mentioned above, KlGal80p could be lost by sequestration to KlGal4p resulting in super-repression when the dissociation rate of the KlGal80-Gal1 complex exceeds a threshold. At dissociation (and association) rates below this threshold but high enough to enable adaptation to changes in Gal1p or Gal80p concentrations, KlGal4p activation would be supported by high a concentration of KlGal1p as evident from the KlGal1p-mediated suppression of super-repression. At a very low dissociation rate, the KlGal1-KlGal80 complex would be in a "frozen" state. In this case, elevated KlGal1p concentration would have no effect. We propose that the latter condition may explain the inability of the KlGal1-209 (N261Y) protein variant to suppress the super-repressed *Klgal80* K5A/R6A mutant. According to the kinetic model any factor with a negative influence on the dynamics of the KlGal1–KlGal80 complex would support KlGal4p activation by reducing sequestration of KlGal80p to KlGal4p. Any factor with a positive influence on the dynamics would run the risk of super-repression, which, however, can be prevented by high KlGal1p levels. The conformational changes associated with the galactokinase catalytic cycle are expected to have a positive influence on the dynamics of a KlGal1-KlGal80 complex. The KlGal1-m1 (E254K) protein, which was described as regulatory deficient and catalytically active may benefit from the higher

dynamics. The reduced affinity for KlGal80p can obviously be compensated by high levels of the KlGal1-m1 (E254K) protein.

The KlGal1-209 (N261) protein can replace wild-type KlGal1p in its function as a co-inducer but this function is impaired by the KlGal80-NLS mutation. The KlGal1-209 (N261) protein is unable to function as a multicopy suppressor in the NLS mutant. In the *KlGAL80* wild-type background, the dynamics of the KlGal1–209–KlGal80 complex might be reduced, first by the stabilizing function of the KlGal80 N terminus and second, by the lack of conformational changes in KlGal1-209p due to the N261Y mutation, which abrogates catalytic activity. The N261Y substitution has the potential to prevent the release of the galactokinase reaction products rather than affecting the catalytic reaction itself (Lavy et al, 2016). The KlGal1-209 protein may therefore be locked in a stable complex with KlGal80p that supports KlGAL4p activation, as mentioned above, but that is nonresponsive to the multicopy suppression effect.

The N261 residue might be required for the complex stabilizing function of the KlGal80 N terminus. N261 is part of the catalytic center of KlGal1p (Fig 7B) and located in the neighbourhood of the proposed docking site for the wild-type N terminus (Fig 7A), which may even allow for a direct interaction between the NLS and N261. Thus, the N261Y mutation might also prevent any rescue of the NLS mutation because the critical residue is not present.

It is tempting to reconcile the structural and the kinetic model on the basis of the overlapping functions of the KlGal80-NLS to an "NLS-model" that has conserved as well as *Kluyveromyces*-specific features. The NLS mediates the nuclear transport of KlGal80p but is not sufficient for the co-transport of KlGal1p and KlGal80p, which requires additional interactions between KlGal80p and KlGal1p via the extended interface. In the NLS model, we propose that the KlGal80-NLS, released from its importin in the nucleus, interacts with the catalytic center of KlGal1p. This interaction inhibits galactokinase activity and catalyzes formation of a (potentially different) KlGal1–KlGal80 complex. Our model implies that the dynamics of the KlGal1–KlGal80 complex must not exceed a threshold, otherwise it runs the risk of super-repression due to sequestration of KlGal80p to KlGal4p. However, the regulation of Gal4 activity by galactose requires that the changing concentrations and affinities can be sensed and transmitted for which a certain dynamics of the Gal80-Gal1/3 complex is a prerequisite. Such a kinetic model differs from the current variants of the dissociation model in an important point: To explain the activation of KlGal4p a structural change in the Gal4–Gal80 complex is not required.

The mechanism of Gal4p activation in *S. cerevisiae* may not be entirely different from that in *K. lactis*. There is evidence that ScGal80p also requires Gal3p and ScGal1p to enter the nucleus (Egriboz et al, 2011), that the dynamics of the Gal80p–Gal3p complex is much lower than the dynamics in the Gal80p–Gal1p complex (Upadhyay, 2014; Lavy et al, 2016), and that a region at the N terminus of ScGal80p is involved in nuclear transport (Nogi & Fukasawa, 1989). However, the overlapping functions of the Gal80 N terminus may have been lost by divergent evolution and perhaps by subfunctionalization of ScGal1p and Gal3p.

The bifunctionality of *K. lactis* Gal1p provides a rational for the different compartment of the regulators in *K. lactis* compared with *S. cerevisiae*. Because interaction between KlGal80p and KlGal1p results in inhibition of the catalytic activity of KlGal1p (Fig 6A; [Anders et al, 2006b]), formation of galactose-1-phosphate (Gal-1P), the first step in galactose metabolism, would be slowed down as KlGal80p

accumulates. However, the overlap of the KlGal80-NLS with the region causing galactokinase inhibition provides a means to restrict enzyme inhibition to the nuclear fraction of KlGal1p. In the cytoplasm binding of the NLS to an importin might occlude the N-terminal KlGal1p interaction site but should still allow the co-transport of KlGal1p and KlGal80p because KlGal1–KlGal80 complex formation can occur via the extended interface homologous to the one revealed in the ScGal3–ScGal80 complex (Lavy et al, 2012). In the nucleus, where dissociation of the importin occurs, the NLS would be released and could find a docking site on KlGal1p. The coupling of KlGal1p nuclear transport to that of KlGal80p seems to be unique and an elegant manifestation of optimization in the simpler setting of the regulatory module in *Kluyveromyces spp.*

## Materials and Methods

### Yeast strains and growth conditions

The *K. lactis* strains (Table S1) JA6/D802 (*gal80-Δ2::ScURA3*) (Zenke et al, 1993), JA6/D1R (*gal1::Scura3*) (Zachariae, 1994), JA6/D1D802R (*gal80-Δ2::Scura3 gal1::Scura3*) (Zachariae, 1994), JA6/G80M (*GAL80*), JA6/G80-KR56A (*gal80-K5A, R6A*), JA6/G80-SV40 (*SV40-NLS-GAL80*), and JA6/G80-SVKR (*SV40-NLS-gal80-K5A, R6A*) were congenic to JA6 (*MAT α ade1-600 adeT-600 ura3-12 trp1-11*) (Breunig & Kuger, 1987) and differed in the indicated alleles. The SV40-NLS (MGAPPKKKRKVA) was fused N-terminally. JA6/G80M, JA6/G80-KR56A, JA6/G80-SV40, and JA6/G80-SVKR contained a C-terminal c-Myc epitope (3×). The different *KlGAL80* alleles were generated by site-directed mutagenesis using the QuikChange Multi Site-Directed Mutagenesis Kit (Agilent Technologies) or by fusion PCR, verified by DNA sequencing, and introduced into JA6/D802 (*Klgal80::ScURA3*) (Zenke et al, 1993) as restriction fragments replacing *Klgal80::ScURA3*. The *S. cerevisiae* strains FI4-s4 ScGAL80 (*ScGAL80*) and FI4-s4 ScGAL80KR56A (*Scgal80-K5A, R6A*) were isogenic to FI4 sin4ΔScgal80Δ (*sin4::HIS5 Scgal80::ScURA3*) (Blueher et al, 2014) except for the indicated *ScGAL80* alleles. The different *GAL80* alleles were generated by site-directed mutagenesis using the QuikChange Multi Site-Directed Mutagenesis Kit (Agilent Technologies) or by fusion PCR, verified by DNA sequencing and introduced into FI4 sin4ΔScgal80Δ as a PCR fragment replacing *Scgal80::ScURA3*. Yeast cells were grown in synthetic minimal medium (0.67% yeast nitrogen base supplied with an amino acid/base mix). Glucose and galactose were added as carbon sources with a final concentration of 2%. For selection of prototroph transformants, the corresponding amino acid or base was left out.

### Plasmids and cloning

The multicopy vector pEG80WTGFPct codes for a C-terminally GFP-fused KlGal80p expressed under control of the ScADH1 promoter (Table S2). The PCR fragment obtained with KlG80SmiIBw (Table S3) and MluATGWTG80fw from pEgal80NLS1GFPct was cut with *Mlu*I and *Smi*I and ligated with the equally cut vector pEgal80NLS1GFPct. The multicopy plasmids pEQRS80A1 - C5 code for GFP-fused KlGal80p fragments expressed the under control of the ScADH1 promoter. The PCR fragments obtained with the corresponding primers (Table

S4) from pEQRS80 (Hager, 2003) were cut with *Mlu*I and *Smi*I and ligated with the equally cut pEQRS80 vector. The multicopy vector pEQRS80DC1 codes for a GFP-fused KlGal80 fragment (residue 40–457) expressed under control of the ScADH1 promoter. The PCR fragment obtained with C2MluIKlG80FwNeu and KlG80SmiIBw from pEQRS80 was cut with *Mlu*I and *Smi*I and ligated with the equally cut vector pEQRS80. The multicopy vector pEgal80NLS1C1 codes for a GFP-fused KlGal80p fragment (residues 2–39) with the amino acid exchange K5A/R6A expressed under control of the *ScADH1* promoter. The PCR fragment obtained with GFPFw and C1SmiIKlG80Bw from pEgal80NLS1 was cut with *Mlu*I and *Smi*I and ligated with the equally cut vector pEQRS80. The multicopy vector pEScG80 codes for a GFP-fused ScGal80p expressed under control of the ScADH1 promoter. The PCR fragment obtained with MluIScGal80fw and SwaIScGal80bw from pScGal80 was cut with *Mlu*I and *Smi*I and ligated with the equally cut vector pEQRS80. The multicopy vector pEScG8036 codes for a GFP-fused KlGal80p fragment (residues 2–36) expressed under control of the *ScADH1* promoter. The PCR fragment obtained with GFPFw and ScG80_36_Smi_Bw from pEScG80 was cut with *Mlu*I and *Smi*I and ligated with the equally cut vector pEQRS80. The multicopy vector pEGFP-KlNT-ScG80 codes for a GFP-fused ScGal80p variant, where the N-terminal 16 amino acids were exchanged with those of KlGal80p, expressed under control of the *ScADH1* promoter. The PCR fragment obtained with KlG80_rv and KlG80NT_15AS_fw from pEScG80 was cut with *Mlu*I and *Smi*I and ligated with the equally cut vector pEQRS80. The multicopy vector pEGFP-Kl56-ScG80 codes for a GFP-fused ScGal80p variant, where the N-terminal 16 amino acids were exchanged with those of KlGal80-K5A/R6Ap, expressed under control of the ScADH1 promoter. The PCR fragment obtained with KlG80_rv and KlG80_KR56A_NT15AS_fw from pEScG80 was cut with *Mlu*I and *Smi*I and ligated with the equally cut vector pEQRS80. pGSTGal1 codes for a glutathione S-transferase-KlGal1 fusion protein Gst-KlGal1p (Zenke et al, 1999). The centromeric plasmid pCGFPAG1 codes for a GFP-KlGAL1 fusion protein expressed under control of the ScADH1 promoter (Anders et al, 2006b). The multicopy vector pEGFPScG80-S8K codes for a GFP-fused ScGal80p variant with the amino acid exchange S8K expressed under control of the *ScADH1* promoter. The PCR fragment obtained with KlG80_rv and ScG80_K8_fw from pEScG80 was cut with *Mlu*I and *Smi*I and ligated with the equally cut vector pEScG80. The Plasmid pCGFPAG1-ura3Δ was obtained by introducing a frameshift mutation into the ScURA3 gene of pCGFPAG1 (Anders et al, 2006b). The centromeric plasmids pCGal1HA and pCNLSGal1HA code for C-terminally HA-tagged KlGal1p without (pCGal1HA) and with an N-terminal SV40-NLS (MGAPPKKKRKVA) (pCNLSGal1HA). The multicopy plasmid pEAG80S2 codes for the variant KlGal80-s2p (E367K) expressed under control of the ScADH1 promoter (Zenke et al, 1999). The multicopy plasmid pEAG80KR56A codes for the *Klgal80* double mutant K5A, R6A under control of the ScADH1 promoter. The *Klgal80-KR56A* containing *Bcu*I and *Pfl23II* fragment of pAG80KR56A was cloned into pEAG80 (Zenke, 1995). pAG80KR56A was constructed by site directed mutagenesis using the primers KlG80KR56A_fw and Amp_DS with pAG80 (Zenke et al, 1999). The plasmid pEAG80-KR56A-SV40 codes for a KlGal80p variant with the exchange K5A/R6A and an N-terminal SV40-NLS-fusion (MGAPPKKKRKVA). pEAG80-KR56A-SV40 was constructed by InFusion cloning of the *Smi*I and *Spe*I cut vector pEAG80 and the PCR products obtained with the primers ADH1-G80-inFu-FW and

ADH1-G80-inFu-RV from pEAG80 as well as G80-KR56A-InFu-FW and G80-KR56A-inFu-RV from pI80KR56ASV40. The integration plasmid pI80KR56ASV40 codes for KlGal80p with the exchange K5A/R6A, C-terminal c-myc tag (3×) and an N-terminal SV40-NLS-fusion (MGAPPKKKRKVA) and was obtained by integration of the *Eco*81I and *Van*91I cut fusion PCR fragment into the same cut vector pI80Myc. The fusion PCR was performed with the primers Integration_pIG80_3 and KlGAL80-422C and the PCR fragments of SV40NLSKlG80_bw and Integration_pIG80_3 as well as SV40NLS_KR56A_fw and KlGAL80-422C with pI80KR56A as the template. The integration plasmid pI80KR56A codes for KlGal80p with the amino acid exchange K5A/R6A and was constructed by ligation of the pKlGal80KR56A *Van*91I and *Eco*81I fragment containing the upstream fragment of *Klgal80-K5A/R6A* with the equally cut pI80Myc. The plasmid pKlGal80KR56A coding for KlGal80p with the exchange K5A/R6A was constructed by site directed mutagenesis with the QuikChange Multi Site-Directed Mutagenesis Kit (Agilent Technologies) with the primers KlGal80NLS1CO and Amp_DS and the plasmid pKlGal80 (Zenke et al, 1993). The integration plasmid pI80Myc coding for c-myc–tagged (3×) KlGal80p was constructed with the QuikChange Multi Site-Directed Mutagenesis Kit, a megaprimer containing the c-myc tag and the plasmid pI80 (Zenke et al, 1993). The megaprimer was obtained by PCR with the primers TagFw and MycTagBw on pYM5 (Knop et al, 1999). Plasmid pETNHG80KR56A is a derivative of pETNHG80 (Anders et al, 2006b) and codes for the N-terminal His$_6$-tagged KlGal80p variant (K5A, R6A, position in the untagged protein). Plasmid pETIHG80KR56A is a derivative of pETIHG80 (Anders et al, 2006b) and codes for the internal His$_6$-tagged KlGal80p variant (K5A, R6A).

## Protein expression and purification

N-terminally or internal 6xHis-tagged wild-type KlGal80p (NHKlGal80p or IHGal80p), double mutant KlGal80-K5A/R6Ap (NHKlGal80-K5A/R6Ap or IHGal80-K5A/R6Ap) and KlGal1p (NHGal1p) were expressed in *E. coli* strain Rosetta (DE3)-pLys from pET-derived (Novagen) expression vectors (pETNHG80, pETIHG80, pETNHG1 [Anders et al, 2006b] pETNHG80-KR56A, and pETIHG80KR56A). Purification was executed as previously described (Anders et al, 2006b). The gene encoding KlGal1p was cloned into pET M30 vector (further referred as GST-Gal1) and pET M11 vector (referred as Gal1). Genes encoding KlGal80p wild-type and K5A/R6A mutant with internal 6xHis tag (amino acids 352–357) were inserted into a pET series vector without any additional tag. KlGal4 gene was cloned into pET M30 vector with a N-terminal 6xHis-GST tag. All proteins were expressed overnight at 18°C in *E. coli* pRARE strain induced with 1 mM IPTG. Bacteria were collected by spinning and stored at −80°C until purification. Pellets were dissolved in 50 mM TRIS–HCl, pH 7.5, 300 mM NaCl, 5% vol/vol glycerol, 10 mM imidazole, 2 mM MgCl$_2$, and 1 mM 2-mercaptoethanol. After protease inhibitors, lysozyme and protease were added, bacteria were lysed by sonication. Lysate was centrifuged at 60,000*g* for 30 min, proteins from supernatant were purified using NiNTA affinity chromatography according to instructions provided by resin manufacturer (QIAGEN). All proteins were eluted in a lysis buffer containing 250 mM imidazole. In case of Gal1 and Gal80 proteins NiNTA eluates were directly applied to Superdex 200 pg HiLoad 26/600 column (GE Healthcare) pre-equilibrated to 20 mM Tris–HCl, 150 mM NaCl, and 2

mM DTT buffer. GST-Gal1 and GST-Gal4 proteins after NiNTA affinity chromatography were purified on a GSTrap HP column (GE Healthcare) with an additional high salt washing step (lysis buffer containing 1 M NaCl). Next, proteins were eluted using 18 mM glutathione (GSH), concentrated and subjected to size exclusion chromatography as mentioned before. After up-concentration, all proteins in a size-exclusion chromatography buffer were snap-frozen in liquid nitrogen and stored at −80°C until use.

## GST pull-down assay

To assess interaction between GST-Gal1 and Gal80 proteins 30 *μ*g of GST-Gal1 and equimolar amount of Gal80 wild-type or K5A/R6A were mixed in a 20 mM TRIS–HCl, pH 7.5, 150 mM NaCl, 2 mM MgCl$_2$, 2 mM DTT, and 0.5% vol/vol Tween 20. To simulate the trigger of GAL regulon, 2% wt/vol galactose and 5 mM ATP were used. In the pull-down that involves Gst-Gal4p, the same ligand concentrations were used before the assay to preoccupy KlGal80p with a threefold molar excess of KlGal1p. Such protein mixtures were incubated at 30°C for 30 min and only then were mixed with GST-Gal4p. Free GST was used as a specificity control. Loading control was prepared before the addition of GSH Sepharose 4B (GE Healthcare). Samples were incubated with beads on a rotating wheel at 4°C for 2 h. GSH beads were collected by centrifugation at 500*g* and washed in corresponding pull-down buffers with or without ligands. After three repetitive washing steps Laemmli sample buffer was added to beads and all bound proteins were denatured at 95°C for 5 min. Proteins were separated using 4–12% gradient precast Bis-Tris Bolt gels (Invitrogen) and stained with Coomassie for imaging.

## Peptide synthesis

The peptide KlN17G80 included the N-terminal 17 residues of KlGal80p (MNNNKRSKLSTVPSSRP) and was synthesized by the group of Prof. Dr. Frank Bordusa, Martin-Luther-University, Halle, Germany.

## Galactokinase inhibition assay

The galactokinase inhibition experiments were performed as described previously (Anders et al, 2006b). Shortly ADP production from ATP in the reaction (galactose + ATP → galactose − 1 − phosphate + ADP) was coupled to the reactions of pyruvate kinase and lactate dehydrogenase monitoring NADH consumption. Each reaction mixture contained 10 nM NHGal1p and increasing concentrations of NHKlGal80p or NHKlGal80-KR56Ap. Samples containing the different KlGal80p variants were measured against a sample with NHKlGal1p alone giving relative galactokinase activities. K$_{D, app}$ values were calculated from the decline in KlGal1p activity with increasing KlGal80p concentrations.

## *β*-galactosidase assay

*β*-galactosidase activity was measured in crude cell extracts or evaluated by visual inspection of colony growth and color on X-Gal plates. For the liquid assay, the substrate for *β*-galactosidase was o-nitrophenyl-*β*-D-galactopyranoside (ONPG). Hydrolysis of ONPG was followed in a buffer containing 5 mM Tris/HCl, pH 7,8; 5% (vol/

vol) glycerol, 10 mM KCl, and 4 mg/ml ONPG, by the increase in light absorbance at 420 nm at 30°C, in a 96 well plate with a plate reader (Spectrostar, Nano Absorbance Reader; BMG LABTECH).

For the plate assay, YNB agar plates contained the indicated carbon source, an amino acid/base mix and the chromogenic substrate 5-bromo-4-chloro-3-indoxyl-$\beta$-D-galactopyranosid (X-Gal) (40 $\mu$g/ml). Cell suspensions of a single colony were serially diluted and spotted on the plates. The plates were incubated at 30°C and documented by scanning after 3 d.

### Thermal shift assay

To compare the thermostability of WT and K5A/R6A Gal80 proteins 10 $\mu$g of corresponding proteins were suspended in 20 mM TRIS–HCl, pH 7.5, 150 mM NaCl$_2$, and 2 mM DTT in a presence of SYPRO Orange (Sigma-Aldrich) hydrophobic fluorescent dye. Samples were gradually heated from 4°C to 98°C at a 0.6°C/min rate. Fluorescence data corresponding to SYPRO Orange emission maximum (ex: 490 nm, em: 583 nm) was collected every 0.2°C. Raw fluorescence data were scaled between 0 and 1. Precise melting points were determined from the first derivative.

### Immunofluorescence microscopy

Immunofluorescence slides were prepared according to the protocol of Atkin (1998). Nuclear staining was performed with DAPI or Hoechst 33342. Primary antibodies were c-Myc (monoclonal, from mouse, by Roche) and c-Myc (polyclonal, from rabbit, by Santa Cruz). Secondary antibody was Alexa Fluor 555 (anti rabbit; Life Technologies). Spheroblasting of cells was done with Zymolyase 100T (Seikagaku). Cells expressing GFP-fused proteins were resuspended in PBS containing 4% formaldehyde and dropped on polyline-coated slides. After 5 min, the slides were washed with PBS, and the cells were coated with 10 $\mu$l Hoechst 33342 (5 $\mu$g/ml) and 10 $\mu$l Mounting Medium (according to Atkin [1998]) and coverslips.

### Structural modelling and docking

Homology-based model of KlGal1p was prepared using SWISS-MODEL server (Waterhouse et al, 2018) with ScGal3 (PDB ID 3V2U) as a template. The obtained model of KlGal1p was combined with the known KlGal80p structure (PDB ID 3E1K) and positioned according to ScGal3p-ScGal80p position in PDB ID 3V2U. The protein–protein interface of the modelled complex was validated using InterProSurf (Negi et al, 2007). PyMOL software was used for visualization purposes (DeLano, 2002). Surface charge distributions were calculated using APBS electrostatics (Jurrus et al, 2018). The KlGal80p 1-16 peptide was placed in five arbitrarily chosen areas of the KlGal1p-KlGal80p model based on surface charge and presence of a potential binding cleft. Prepared models for WT KlGal80 1-16 and K5A/R6A mutant peptide were submitted to FlexPepDock server (Raveh et al, 2010; London et al, 2011). Movements within the KlGal1–KlGal80 complex were restrained, whereas the KlGal80p 1-16 peptide could undergo up to 10 Å of movement. 100 low-resolution models and 100 high-resolution models were prepared for of each position. The 10 models with lowest Rosetta scores (which indicates highest binding probability) were collected.

# Supplementary Information

# Acknowledgements

We thank Katharina Böhm for construction of the pEGFPScG80-S8K plasmid and Renate Langhammer for fruitful discussions. This work was supported by grant BR921/7 and BR921/9-1 (KD Breunig and A Reinhardt-Tews) from the Deutsche Forschungsgemeinschaft (DFG). In addition, R Krutyhołowa and S Glatt were supported by grant OPUS16 2018/31/B/NZ1/03559 from the Polish National Science Centre. We thank the MCB structural biology core facility (supported by the TEAM TECH CORE FACILITY/2017-4/6 grant from Foundation for Polish Science) for providing computational resources.

## Author Contributions

A Reinhardt-Tews: data curation, validation, investigation, and visualization.
R Krutyhołowa: data curation, formal analysis, visualization, and methodology.
C Günzel: formal analysis and methodology.
C Roehl: investigation and methodology.
S Glatt: conceptualization, software, formal analysis, funding acquisition, validation, investigation, and methodology.
KD Breunig: conceptualization, data curation, formal analysis, supervision, funding acquisition, project administration, and writing—review and editing.

## Conflict of Interest Statement

The authors declare that they have no conflict of interest.

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
