## [Reviewer comments · Life Science Alliance]

Life Science Alliance

A double role of the Gal80 N-terminus in activation of transcription by Gal4p

Annekathrin Reinhardt-Tews, Rościsław Krutyhołowa, Christian Günzel, Constance Roehl, Sebastian Glatt, and Karin Breunig

DOI: <https://doi.org/10.26508/lsa.202000665>

Corresponding author(s): Karin Breunig, Serymun Yeast GmbH and Sebastian Glatt, Jagiellonian University, Malopolska Centre of Biotechnology

Review Timeline:

Submission Date:	2020-02-01
Editorial Decision:	2020-02-25
Revision Received:	2020-09-02
Editorial Decision:	2020-09-21
Revision Received:	2020-09-29
Accepted:	2020-09-29

Scientific Editor: Shachi Bhatt

Transaction Report:

February 25, 2020

Re: Life Science Alliance manuscript #LSA-2020-00665

Prof. Karin D. Breunig
Martin Luther University Halle-Wittenberg
Institute of Biology
Weinbergweg 10
Halle 06120
Germany

Dear Dr. Breunig,

Thank you for submitting your manuscript entitled "A double role of the Gal80 N-terminus in activation of transcription by Gal4p" to Life Science Alliance. The manuscript was assessed by expert reviewers, whose comments are appended to this letter.

As you will see, the reviewers appreciate your conclusions. However, they both think that more support is needed for the major claims made, and they provide constructive input on how to provide such support. We would thus like to invite you to submit a revised version of your manuscript, addressing the individual points raised by the reviewers. Please also make sure that the organism you are working on is easily recognizable from the abstract and that the yeast cells analyzed are healthy (eg, we are worried by the images shown in Fig 1 panels E/F and Sup figure S2).

Thank you for this interesting contribution to Life Science Alliance. We are looking forward to receiving your revised manuscript.

Sincerely,

B. MANUSCRIPT ORGANIZATION AND FORMATTING:

Reviewer #1 (Comments to the Authors (Required)):

The authors have identified an NLS at the N-terminus of KIGal80, the inhibitor of KIGal4. The authors have fused KIGal80 to GFP and found that while wild-type Gal80 localized to the nucleus,

Gal80 with the mutated NLS K5A/R6A was not localized to the nucleus but present everywhere inside the cell. One would expect that the KGal80K5A/R6A mutant strain was deficient for inhibition of KGal4, however, the authors have found that it is actually deficient for induction, as the presence of the mutant KGal80K5A/R6A prevented activation of beta-galactosidase under both repressing (glucose) and inducing conditions (galactose). The authors have found that the over-expression of Gal1 restores the induction of beta-galactosidase in the KGal80K5A/R6A mutant strain and the authors have shown that the K5A/R6A mutation reduces the interaction of KGal80 with KGal1. The authors conclude that the NLS serves two functions: it localizes KGal80 to the nucleus, where it binds and inhibits KGal4, and it interacts with KGal1, which serves to switch the KGal80-KGal4 complex to the KGal80-KGal1 complex, setting KGal4 free to activate transcription. This is an interesting article which should be published in Life Science Alliance after the criticisms below have been addressed.

Minor criticism:

According to the materials and methods, all fusions expressed from plasmids are under the control of the *S. cerevisiae* ADH1 promoter. While it would have been better to use the KGal80 promoter, the fluorescent microscopic pictures presumably look nicer this way. It is not clear if the KGal80-myc fusions were integrated such that they are expressed under the control of the KGal80 promoter or also under the control of the ScADH1 promoter. In order to be more transparent, the authors should state for each construct from which promoter it is expressed in the respective figure legends.

Major criticisms:

The inhibitory effect of the KGal80K5A/R6A mutant could be due to high protein levels. For the KGal80-myc3 and KGal80K5A/R6A-myc3 fusions, the authors should run a Western Blot with cells grown under repressing and inducing conditions in order to determine the relative expression levels.

The authors have only shown that KGal1 interacts better with wild-type KGal80 than with the KGal80K5A/R6A mutant protein. They should set up a competition assay to directly demonstrate their claim that KGal1 competes better for the wild-type KGal80-KGal4 complex as compared to the mutant KGal80K5A/R6A-KGal4 complex.

Reviewer #2 (Comments to the Authors (Required)):

The core features of GAL regulation show strong conservation, however, species-specific differences in regulation are known to arise due to differences in expression levels, localization and primary sequence of the regulatory proteins: Gal4, Gal80 and Gal1/Gal3. Reinhardt-Tews et al. show that the N terminal 16 amino acids of Gal80 (Gal80-N16AA) are required for both nuclear localization and Gal1 binding in *K.lactis* but not in *S.cerevisiae*. The authors recapitulate their previous finding that Gal80 is predominantly nuclear in *K.lactis*, but not in *S.cerevisiae*, and utilize their previously established binding, expression and localization assays to place the role of Gal80-N16AA in the overall GAL regulation. The main/novel claim of this study is that Gal80-N16AA interacts with the catalytic center of nuclear Gal1, which in turn destabilizes Gal80-Gal4 interaction. This is a well written manuscript with clear results that propose a potentially interesting mechanism. However, the main claim of the manuscript needs a more thorough validation.

Major concern

The claim for the interaction of Gal80-N16AA with the catalytic site of Gal1 is supported only

through in silico model and not the actual structure. It is clear this region is necessary but not sufficient for interaction with Gal1. Thus, mutations in Gal1 that disrupts interaction with Gal80-N16AA, based on their model, are essential to support their claim.

Minor concerns

1. The suppression of Gal80-KR56A mutation by Gal1 mutant that does not interact with wild-type GAL80 but no suppression by the Gal1 catalytic mutant is a non-intuitive result and might be remotely related to the claim of this paper. This should be discussed.
2. Typo on Page 10 "color on 0,2% glucose"
3. In Figure 7 the models are hard to follow

Author Responses Point-by-Point**Reviewer #1:**

The authors have identified an NLS at the N-terminus of KIGal80, the inhibitor of KIGal4. The authors have fused KIGal80 to GFP and found that while wild-type Gal80 localized to the nucleus, Gal80 with the mutated NLS K5A/R6A was not localized to the nucleus but present everywhere inside the cell. One would expect that the KIGal80K5A/R6A mutant strain was deficient for inhibition of KIGal4, however, the authors have found that it is actually deficient for induction, as the presence of the mutant KIGal80K5A/R6A prevented activation of beta-galactosidase under both repressing (glucose) and inducing conditions (galactose). The authors have found that the over-expression of Gal1 restores the induction of beta-galactosidase in the KIGal80K5A/R6A mutant strain and the authors have shown that the K5A/R6A mutation reduces the interaction of KIGal80 with KIGal1. The authors conclude that the NLS serves two functions: it localizes KIGal80 to the nucleus, where it binds and inhibits KIGal4, and it interacts with KIGal1, which serves to switch the KIGal80-KIGal4 complex to the KIGal80-KIGal1 complex, setting KIGal4 free to activate transcription. This is an interesting article which should be published in Life Science Alliance after the criticisms below have been addressed.

Minor criticism:

According to the materials and methods, all fusions expressed from plasmids are under the control of the *S. cerevisiae* ADH1 promoter. While it would have been better to use the KIGAL80 promoter, the fluorescent microscopic pictures presumably look nicer this way. It is not clear if the KIGal80-myc fusions were integrated such that they are expressed under the control of the KIGAL80 promoter or also under the control of the ScADH1 promoter. In order to be more transparent, the authors should state for each construct from which promoter it is expressed in the respective figure legends.

Authors Answer: We have used the *S. cerevisiae* ADH1 to express the GFP-fusions for several experiments requiring comparable amounts of mutant and wild-type Gal80 protein (e.g. in the NLS-mapping experiment) The KIGAL80 promoter is autoregulated and weak under non-inducing conditions. As suggested the promoter driving expression is now given in the legend and in the plasmid list (Suppl. Table 2).

We would also like to highlight that the yeast strains expressing 3x-myc tagged versions have been fixed and permeabilized during the staining procedure to allow the respective antibody to enter the cells. Therefore, the cells in the light microscopy image look miserable whereas the cells expressing GFP-Gal80 look healthy. We have now marked the two techniques, immunofluorescence and (GFP) fluorescence, by red and green bars in a new version of Figure 1. In addition, we have modified the

figure legends of Figures 1, 2, 4 and S1 to S3 to point out more clearly which technique was used in the respective experiments.

Major criticisms:

The inhibitory effect of the KIGal80K5A/R6A mutant could be due to high protein levels. For the KIGal80-myc3 and KIGal80K5A/R6A-myc3 fusions, the authors should run a Western Blot with cells grown under repressing and inducing conditions in order to determine the relative expression levels.

Authors Answer: We show in the revised version by a Western blot (new Fig. 3B) that wild-type and mutant KIGal80p are expressed at comparable levels. Therefore, we can exclude the possibility that the KIGal80K5A/R6A mutant exhibits its inhibitory effect because of elevated protein levels. We now refer to this new data on page 7, which reads as follows “We can exclude that this phenotype is caused by an elevated concentration of the KIGal80-K5A/R6A protein, the Western blot indicated no difference to wild-type KIGal80p (**Fig. 3B**)”.

In addition, we have optimized the purification of KIGal80 and KIGal80K5A/R6A from bacteria and measured the respective thermo-stability. Both, wt and mutant proteins show almost identical melting profiles, indicating that the mutation has no influence on the stability of the KIGal80 protein. We now show these data in the newly created Suppl Fig S5 and mention the data on page 13, which now reads as follows – “Since the mutated KIGal80-K5A/R6Ap variant had the same thermostability as wild-type KIGal80p (**Supplementary Fig. S5**) we conclude that the KR>AA exchange is responsible for the reduced affinity of KIGal80-K5A/R6Ap for GstKIGal1p *in vitro*.

Major criticisms:

The authors have only shown that KIGal1 interacts better with wild-type KIGal80 than with the KIGal80K5A/R6A mutant protein. They should set up a competition assay to directly demonstrate their claim that KIGal1 competes better for the wild-type KIGal80-KIGal4 complex as compared to the mutant KIGal80K5A/R6A-KIGal4 complex.

Authors Answer: We agree with the reviewer that such *in vitro* competition experiments with purified proteins would have been the most mechanistic evidence of the proposed model. It is indeed important to know whether there is an affinity difference for KIGal1p between the KIGal4p-KIGal80p and the KIGal4p-KIGal80K5A/R6A complexes. We followed the recommendation and attempted to perform the envisioned experiments. These experiments took quite some time, since we were heading for purified complexes, which required several new constructs, establishment of (co)purification

schemes, identification of reconstitution conditions and setting up numerous experiments of which only two are shown in the new Figs 6C and D.

In summary, we were able to form and purify the KIGal1-KIGal80 and KIGal4-KIGal80 complexes *in vitro* with both variants of KIGal80, namely WT and K5A/R6A mutant. Nonetheless, we were not able to purify Gal4p alone despite several attempts and the addition of solubility tags did not change this outcome.

Therefore, it was technically not feasible to perform the competition experiment – titration of Gal4p against the preformed wt and mutant Gal1-Gal80 complexes. Using optimized purification protocols for untagged KIGal80p, we were able to confirm the drastically reduced affinity of the K5A/R6A mutant protein for KIGal1p in the presence of Gal and ATP. As mentioned above, the wild type and mutant Gal80 proteins show identical stability profiles and the inability of KIGal80K5A/R6A to bind Gal1p (in the presence of ATP and galactose) is not related to a general destabilization of the protein (see answer above). We now show this data in the new Figure 6C, replacing the previous version (corresponding input controls are shown in the newly created Supplementary Fig. S6). In the new Fig. 6D we have addressed the question raised by the reviewer in a modified form as described in the text on page 13, which reads as follows “Both KIGal80p variants were detected in the GstGal4p pull down fraction and the intensity of the bands representing one or the other KIGal80p variant was neither affected by the presence or absence of galactose and ATP nor by the presence or absence of KIGal1p in the assay (**Fig. 6D**). Hence the amount of KIGal80p in the bound fraction was only determined by the GstKIGal4p input (**Supplementary Fig. S6**). We conclude that in this three-component *in vitro* binding assay KIGal1p cannot effectively compete with GstKIGal4p.”

Furthermore, we managed to obtain GST-KIGal4-Gal80 complexes using wild type and mutated KIGal80. We have used these preparations to titrate large excess of purified untagged KIGal1 protein, but we have not obtained any evidence for a displacement of a formed KIGal4-KIGal80 complex under any of the tested conditions. We believe that *in vitro* large differences in affinity exist between the KIGal80-Gal4 (very high affinity) and the KIGal80-KIGal1p (low affinity). These large differences do not allow the displacement of wild type KIGal80 from the Gal4 complex by KIGal1. Therefore, we were not able to test the differences between wild type and mutated KIGal80 in this *in vitro* setup or to exclude that these differences do not exist. Most importantly we have shown that in this *in vivo* scenario the basic residues in the N-terminus have a major impact on KIGal4 activation. In a modification of our model we are taking into account the new *in vitro* data and assume that the mutation has no major influence on the KIGal80-KIGal4 complex. To better explain our kinetic model, which for the time being has to remain of hypothetical nature, the discussion section was extensively rewritten and adjusted.

Reviewer #2:

The core features of GAL regulation show strong conservation, however, species-specific differences in regulation are known to arise due to differences in expression levels, localization and primary sequence of the regulatory proteins: Gal4, Gal80 and Gal1/Gal3. Reinhardt-Tews et al. show that the N terminal 16 amino acids of Gal80 (Gal80-N16AA) are required for both nuclear localization and Gal1 binding in *K. lactis* but not in *S. cerevisiae*. The authors recapitulate their previous finding that Gal80 is predominantly nuclear in *K. lactis*, but not in *S. cerevisiae*, and utilize their previously established binding, expression and localization assays to place the role of Gal80-N16AA in the overall GAL regulation. The main/novel claim of this study is that Gal80-N16AA interacts with the catalytic center of nuclear Gal1, which in turn destabilizes Gal80-Gal4 interaction. This is a well written manuscript with clear results that propose a potentially interesting mechanism. However, the main claim of the manuscript needs a more thorough validation.

Major concern

The claim for the interaction of Gal80-N16AA with the catalytic site of Gal1 is supported only through *in silico* model and not the actual structure. It is clear this region is necessary but not sufficient for interaction with Gal1. Thus, mutations in Gal1 that disrupts interaction with Gal80-N16AA, based on their model, are essential to support their claim.

Authors Answer: If the mutation disrupts a specific amino acid interaction of N16AA with KIGal1p that is necessary for KIGal4p activation, there may be *KlGal1* mutants that have a suppressor phenotype because the interaction is restored. We have tried to obtain such mutants by mutagenesis of *KIGAL1* screening for mutants that suppress the KIGAL80-K5A/R6A mutation. We have obtained and further characterized 8 such mutants. All but one had multiple (2-4) amino acid exchanges, three of them had identical double mutations. Six of the amino acid substitutions were introduced as single mutations into the *KIGAL1* WT gene. Among them two are of major interest by affecting specifically the *KlGal80-K5A/R6A* mutant but not the *KIGAL80* WT allele. We provide a confidential cartoon where these residues are marked, specifically for the reviewers. For us, the positions of the mutations did not reveal any candidate with the desired properties. Hence, we decided not to include these data in the present manuscript. A further characterization of the mutants is for sure required, but beyond the scope of this report.

Minor concerns

1. The suppression of Gal80-KR56A mutation by Gal1 mutant that does not interact with wild-type GAL80 but no suppression by the Gal1 catalytic mutant is a non-intuitive result and might be remotely related to the claim of this paper. This should be discussed.

Authors Answer: We thank the reviewer for this stimulating comment. We had a closer look at the *Kgal1* mutants that had been described and came up with a kinetic model to explain their various phenotypes. Intriguingly, the proposed mode of regulation would also explain a number of findings in the literature that cannot easily be reconciled with the current dissociation models. This model is supported by the new biochemical data shown in the new version of Figure 6D and the genetic data shown in Figure 5. These data are discussed in the light of the new kinetic model in the revised version of the manuscript. The discussion section has therefore been revised extensively by reordering paragraphs and extending or replacing some sections. The individual changes have not been marked in the marked version of the manuscript. Instead the Discussion section is marked as new in its entirety.

2. Typo on Page 10 "color on 0,2% glucose"

Authors Answer: The typo has been corrected

3. In Figure 7 the models are hard to follow

Authors Answer: We have removed figure 7 and replaced it by a detailed representation of the postulated secondary interaction between the KGal80 N-terminus and the catalytic center of KGal1p.

September 21, 2020

RE: Life Science Alliance Manuscript #LSA-2020-00665R

Prof. Karin D. Breunig
Martin Luther University Halle-Wittenberg
Institute of Biology
Weinbergweg 10
Halle 06120
Germany

Dear Dr. Breunig,

Thank you for submitting your revised manuscript entitled "A double role of the Gal80 N-terminus in activation of transcription by Gal4p". We would be happy to publish your paper in Life Science Alliance (LSA) pending final revisions necessary to meet our formatting guidelines.

Please make the following edits to comply with LSA's formatting guidelines,

- please upload your tables as editable doc or excel files
- please add a callout in your main manuscript text for Figure 1A, Figure 2A,B and Figure S3

A. FINAL FILES:

-- Summary blurb (enter in submission system): A short text summarizing in a single sentence the study (max. 200 characters including spaces). This text is used in conjunction with the titles of papers, hence should be informative and complementary to the title. It should describe the context and significance of the findings for a general readership; it should be written in the present tense

and refer to the work in the third person. Author names should not be mentioned.

B. MANUSCRIPT ORGANIZATION AND FORMATTING:

Sincerely,

Shachi Bhatt, Ph.D.
Executive Editor
Life Science Alliance

Reviewer #1 (Comments to the Authors (Required)):

The authors have identified an NLS at the N-terminus of KIGal80, the inhibitor of KIGal4. The authors have fused KIGal80 to GFP and found that while wild-type Gal80 localized to the nucleus, Gal80 with the mutated NLS K5A/R6A was not localized to the nucleus but present everywhere inside the cell. One would expect that the KIGal80K5A/R6A mutant strain was deficient for inhibition of KIGal4, however, the authors have found that it is actually deficient for induction, as the

presence of the mutant KIGal80K5A/R6A prevented activation of beta-galactosidase under both repressing (glucose) and inducing conditions (galactose). The authors have found that the over-expression of Gal1 restores the induction of beta-galactosidase in the KIGal80K5A/R6A mutant strain and the authors have shown that the K5A/R6A mutation reduces the interaction of KIGal80 with KIGal1. The authors conclude that the NLS serves two functions: it localizes KIGal80 to the nucleus, where it binds and inhibits KIGal4, and it interacts with KIGal1, which serves to switch the KIGal80-KIGal4 complex to the KIGal80-KIGal1 complex, setting KIGal4 free to activate transcription. This is an interesting article which can now be published in Life Science Alliance as the authors have been able to address the concerns that I had raised for the originally submitted manuscript.

Reviewer #2 (Comments to the Authors (Required)):

The authors have made significant changes and provided convincing evidences to support their claims. This has further strengthened a very good study.

September 29, 2020

RE: Life Science Alliance Manuscript #LSA-2020-00665RR

Prof. Karin D. Breunig
serymun Yeast GmbH
Weinbergweg 22
Halle 06120
Germany

Dear Dr. Breunig,

Thank you for submitting your Research Article entitled "A double role of the Gal80 N-terminus in activation of transcription by Gal4p". It is a pleasure to let you know that your manuscript is now accepted for publication in Life Science Alliance. Congratulations on this interesting work.

The callout for the Figure S3 is still missing from the manuscript text, but you should be able to add it at the proofs stage.

*****IMPORTANT:** If you will be unreachable at any time, please provide us with the email address of an alternate author. Failure to respond to routine queries may lead to unavoidable delays in publication.*******

DISTRIBUTION OF MATERIALS:

Again, congratulations on a very nice paper. I hope you found the review process to be constructive and are pleased with how the manuscript was handled editorially. We look forward to future exciting

submissions from your lab.

Sincerely,

Shachi Bhatt, Ph.D.
Executive Editor
Life Science Alliance